:ۻ: PLOS | ONE

# GluD1 knockout mice with a pure C57BL/6N background show impaired fear memory, social interaction, and enhanced depressive-like behavior

Chihiro Nakamoto[1,2,3], Meiko Kawamura[1], Ena Nakatsukasa[1], Rie Natsume[1], Keizo Takao[4,5], Masahiko Watanabe[6], Manabu Abe[1]*, Tomonori Takeuchi🆔[2,3]*, Kenji Sakimura[1]

1 Department of Animal Model Development, Brain Research Institute, Niigata University, Niigata, Japan, 2 Department of Biomedicine, Aarhus University, Aarhus, Denmark, 3 Danish Research Institute of Translational Neuroscience–DANDRITE, Nordic-EMBL Partnership for Molecular Medicine, Aarhus University, Aarhus, Denmark, 4 Graduate School of Innovative Life Science, University of Toyama, Toyama, Japan, 5 Life Science Research Center, University of Toyama, Toyama, Japan, 6 Department of Anatomy, Faculty of Medicine, Hokkaido University, Sapporo, Japan

* tomonori.takeuchi@biomed.au.dk (TT); manabu@bri.niigata-u.ac.jp (MA)

**Data Availability Statement:** •Raw data of western blot: https://doi.org/10.6084/m9.figshare.

## Abstract

The GluD1 gene is associated with susceptibility for schizophrenia, autism, depression, and bipolar disorder. However, the function of GluD1 and how it is involved in these conditions remain elusive. In this study, we generated a *Grid1* gene-knockout (GluD1-KO) mouse line with a pure C57BL/6N genetic background and performed several behavioral analyses. Compared to a control group, GluD1-KO mice showed no significant anxiety-related behavioral differences, evaluated using behavior in an open field, elevated plus maze, a light-dark transition test, the resident-intruder test of aggression and sensorimotor gating evaluated by the prepulse inhibition test. However, GluD1-KO mice showed (1) higher locomotor activity in the open field, (2) decreased sociability and social novelty preference in the three-chambered social interaction test, (3) impaired memory in contextual, but not cued fear conditioning tests, and (4) enhanced depressive-like behavior in a forced swim test. Pharmacological studies revealed that enhanced depressive-like behavior in GluD1-KO mice was restored by the serotonin reuptake inhibitors imipramine and fluoxetine, but not the norepinephrine transporter inhibitor desipramine. In addition, biochemical analysis revealed no significant difference in protein expression levels, such as other glutamate receptors in the synaptosome and postsynaptic densities prepared from the frontal cortex and the hippocampus. These results suggest that GluD1 plays critical roles in fear memory, sociability, and depressive-like behavior.

10053092.v1 •Raw data of behavior analysis:
https://doi.org/10.6084/m9.figshare.10052663.v1

**Funding:** Funded by (TT) 1) NNF17OC0026774;
Novo Nordisk Foundation Young Investigator
Award 2017; https://novonordiskfonden.dk/en/. 2)
75451; Aarhus Institute of Advanced Studies
(AIAS)-EU FP7 Cofund programme; http://aias.au.
dk/. 3) DANDRITE-R248-2016-2518;
Lundbeckfonden; https://www.lundbeckfonden.
com/en/. (KS) 1) 16H04650; the Ministry of
Education, Culture, Sports, Science and
Technology of Japan (MEXT); http://www.mext.go.
jp/en/. 2) 16H06276; the Ministry of Education,
Culture, Sports, Science and Technology of Japan
(MEXT); http://www.mext.go.jp/en/.

**Competing interests:** The authors have declared
that no competing interests exist.

## Introduction

The δ-type ionotropic glutamate receptor consists of GluD1 (GluRδ1) and GluD2 (GluRδ2) [1–3]. Despite having conserved membrane topology and amino acid residues critical for glutamate binding and $Ca^{2+}$ permeability, the δ subfamily members do not function as conventional glutamate-gated receptor channels when expressed alone or in combinations with other ionotropic glutamate receptor subunits [4–6]. Instead, they are components of a tripartite transsynaptic adhesion system, where the extracellular domain of postsynaptic GluD1/2 interacts with that of presynaptic neurexin protein (NRXN) via members of the cerebellin precursor protein (CBLN) family in the synaptic cleft [7–9]. Moreover, slow activity of GluD1/2 ion channels is triggered by activation of group 1 metabotropic glutamate receptors (mGluRs) [10–12].

GluD2 has been intensively studied since it was cloned in 1993 [2,3]. In the rodent cerebellum, GluD2 is exclusively expressed in Purkinje cells and is selectively localized to postsynaptic spines at parallel fiber synapses [13,14]. GluD2 is indispensable during the formation and maintenance of parallel fiber-Purkinje cell synapses by interaction with presynaptic neurexins through its amino-terminal domain [7,8,15,16]. In addition, GluD2 regulates cerebellar synaptic plasticity [15,17] and motor learning [15,18,19] by interaction with the scaffolding proteins through its carboxyl-terminal domain [20–22].

The gene encoding GluD1 in humans (*GRID1*) is associated with a susceptibility for schizophrenia [23–26], major depressive disorder [27], and autism spectrum disorder [28–31]. GluD1 is expressed in various brain regions in rodents, including the cerebral cortex, hippocampus, amygdala, bed nucleus of the stria terminalis, striatum, thalamus, nucleus accumbens, lateral habenular, and the dorsal raphe nucleus [3,32,33]. Similar to GluD2, postsynaptic GluD1 is required for synapse formation and maintenance *in vitro* via CBLN1/CBLN2/CBLN4 and presynaptic NRXN [7,9,34–37]. In addition, GluD1 regulates group 1 mGluRs-mediated long-term depression in the hippocampus *ex vivo* [38]. Furthermore, activation of group 1 mGluRs trigger the opening of GluD1 channels, which are key determinants of the slow excitatory postsynaptic current *ex vivo* [12].

The GluD1 knockout (GluD1-KO) mice line (*Grid1*^tm1Jnz), compared to the relevant control groups, showed abnormal behavioral phenotypes, such as hyper locomotor activity, lower anxiety-like behavior, hyper aggression, higher depression-like behavior, deficits in social interaction [39], enhanced working memory, deficit in contextual and cued fear conditioning [40], and increased stereotyped behavior [41]. However, the *Grid1*^tm1Jnz mouse was generated from embryonic stem (ES) cells derived from the 129/SvEv strain, followed by backcrossing to C57BL/6 mice 2–6 times [39,40,42]. It is well known that backcrossing with different mouse strains leads to a change in the basal levels of behaviors, such as anxiety [43–45], aggression [46,47], prepulse inhibition [48], social interaction [49,50], pain sensitivity [51], depressive-like behavior [44,52–54], and learning and memory [43,55–58]. Thus, a concern of using a mixed genetic background is that the resulting phenotype cannot be confidently attributed to either the target gene or closely-linked genes flanking the targeted locus [55,59,60]. In addition, the 129S6/SvEvTac strain used in *Grid1*^tm1Jnz lacked the DISC1 gene (disrupted in schizophrenia 1) [61–63], which is a strong candidate gene that contributes to cause schizophrenia and autism spectrum disorder [64,65]. Moreover, the *Grid1*^tm1Jnz mouse was generated by knock-in of a selection marker of neomycin cassette; the promoter of the cassette unexpectedly affected gene expression levels [66–69].

To avoid these issues, we generated GluD1-KO mice with a pure C57BL/6N genetic background and investigated an impact of GluD1 deletion on various behaviors including anxiety, aggression, sensorimotor gating, sociability, learning and memory, and depression.

## Methods

### Animals

GluD1-KO mice were generated using the C57BL/6N ES cell line, RENKA [67] and maintained in a pure C57BL/6N background [70]. Briefly, exon 4 of the *Grid1* gene and a *Pgk* promoter-driven neomycin-resistance cassette were flanked by loxP sequences (*Grid1*flox). *Grid1*flox mice were crossed with telencephalin-Cre mice [71] to create the null allele (*Grid1*−). Mice were fed *ad libitum* with standard laboratory chow (Oriental NMF, Oriental Yeast Co., Tokyo) and water in standard animal cages in a 12-h light/dark cycle (light on at 8:00 a.m.) at room temperature and relative humidity in the ranges of 22°C–24°C and 30%–70%, respectively. Experimental protocols used throughout the study were approved by an institutional committee at Niigata University (SA00466) and were in accord with Japanese legislation concerning animal experiments.

Behavioral tests were carried out with 8 to 12-week-old male wild-type (WT, *Grid1*+/+) (n = 115 in total) and GluD1-KO (*Grid1*−/−) (n = 92 in total) litter mates by heterozygous breeding, and were performed during the light phase (between 10:00 a.m. and 18:00 p.m.). Mice were handled (3 min per day for 3 days) before starting behavioral tests. Numbers of animals required was based on previous reports of *Grid1*tm1Jnz mice [39–41]. Behavioral analyses were performed with the experimenter blind to mice genotype. After each trial, the apparatus was cleaned with hypochlorous water to prevent a bias due to olfactory cues. A battery of behavioral tests were performed in the following order: open field, light-dark transition, elevated plus maze, 3-chamber social interaction, and a forced swim.

### Open field

Open field tests were carried out using a method similar to that reported previously, with minor modification [72]. Each mouse was placed in the corner of an open field apparatus (50 cm × 50 cm × 40 cm high; O'Hara & Co., Tokyo, Japan) with a chamber illuminated at either 5 or 100 lux. Distance traveled and time spent in the central area (defined as 25% of total area) were recorded and calculated automatically over a 10-min period using Image OFCR software (O'Hara & Co.; see 'Image analysis for behavioral tests').

### Elevated plus maze

Elevated plus maze tests were carried out using a method similar to that reported previously, with minor modification [72]. The apparatus consisted of two open arms (25 cm × 5 cm) and two enclosed arms of the same size with transparent walls (height 15 cm). The arms and central square (5 cm × 5 cm) were made of white plastic plates and were elevated 60 cm above the floor (O'Hara & Co.). Arms of the same type were oriented opposite from each other. Each mouse was placed in the central square of the maze, facing one of the closed arms. The time spent in closed and open arms and the frequency of entry into open arms were observed for 10 min under two different illumination condition (5 and 100 lux). Data acquisition was performed automatically using Image EP software (O'Hara & Co.; see 'Image analysis for behavioral tests').

### Light-dark transition test

Light-dark transition tests were carried out using a method similar to that reported previously, with minor modification [72]. The apparatus consisted of a cage (21 cm × 42 cm × 25 cm high) divided into 2 equal chambers by a black partition containing a small opening (5 cm x 3 cm high) (O'Hara & Co.). One chamber was made of white plastic and was brightly

illuminated (252 lux), whereas the other chamber was made of black plastic and was dark (no illumination). Mice were placed in the dark chamber and allowed to move freely between the two chambers for 10 min. Time spent in each chamber, total number of transitions and latency to the first transition from dark to light chambers were recorded automatically using Image LD software (O'Hara & Co.; see 'Image analysis for behavioral tests').

### Resident-intruder test

Resident-intruder tests were carried out using a method similar to that reported previously [39]. Resident male WT (27.3 ± 0.2 g) or GluD1-KO mice (24.1 ± 0.4 g) were individually housed for 3–4 weeks before testing. Resident mice were exposed to intruder male WT C57BL/6 mice, which had been group-housed (four to five per cage) and were of lower body-weight than resident mice (0–4 g lighter than intruder mice), for a duration of 10 min. New intruder mice were used in each test. Latency to attack the intruder and attack frequency were measured manually.

### Prepulse inhibition test

Acoustic startle response and prepulse inhibition (PPI) of the acoustic startle response were measured using a startle chamber (SR-Lab Systems; San Diego Instruments, CA, USA) [73]. For acoustic startle responses, a background of white noise was used (70 db). An animal was placed in the Plexiglass cylinder and each test session began after 5 min of acclimatization. Mice were presented with 64 trials. There were eight different sound levels presented: 75, 80, 85, 90, 95, 100, 110, and 120 dB. Each white-noise stimulus was 40 ms and presented 8 times in a pseudorandom order such that each sound level was presented within a block of 8 trials. The intertrial interval was 15 s. Analysis for startle amplitudes was based on the mean of the seven trials (ignoring the first trial) for each trial type.

PPI responses were measured with acoustic stimuli (120 dB) combined with four different prepulse intensities. Each mouse was placed in the startle chamber and initially acclimatized for 5 min with background white noise alone (70 dB). Mice were then presented with 48 trials. Each session consisted of six trial types. One trial type used a sound burst (40 ms, 120 dB) as the startle stimulus (startle trials). There were four different trials consisting of acoustic pre-pulse and acoustic startle stimuli (prepulse trials). The prepulse stimulus (20 ms) of either 73, 76, 79, or 82 dB was presented 100 ms before the onset of the acoustic startle stimulus. Finally, there were trials where no stimulus was presented (no-stimulus trials). The six trial types were presented in a pseudorandom order such that each trial type was presented once within a block of eight trials. The intertrial interval was 15 s. Analysis was based on the mean of the seven trials (ignoring the first trial) for each trial type. The percentage PPI of a startle response was calculated using the following equation: $100 - [100 \times (\text{startle response on prepulse trials} - \text{no stimulus trials})/(\text{startle trials} - \text{no stimulus trials})]$.

### Three-chambered social interaction test

The three-chambered social interaction test was performed as previously described, with minor modification [50,72]. The apparatus consisted of a rectangular, illuminated (5 lux) three-chambered box with a lid and an attached infrared video camera (O'Hara & Co.). Each chamber was 20 cm × 40 cm × 22 cm (high) and the dividing walls were made of clear Plexiglas, with small square openings (5 cm wide × 3 cm high) to allow exploration of each chamber. Male mice of the C3H strain, with ages ranging between 8 to 12 weeks, were purchased from Charles River Laboratories (Yokohama, Japan) and used as 'strangers'.

One day before testing, the 'subject mice' were individually placed in the middle chamber and allowed to freely explore the entire apparatus for 5 min. Before testing, subject mice were placed in the middle chamber and allowed to freely explore all three chambers for 10 min (habituation trial). In the sociability test (sociability trial), an unfamiliar C3H male mouse ('stranger 1') that had no prior contact with the subject mouse was placed in one of the side chambers. The stranger mouse was enclosed in a small, round wire cage, which allowed nose contact between the bars but prevented fighting. This cage was 11 cm in height, with a floor diameter of 9 cm and vertical bars 0.5 cm apart. The subject mouse was placed in the middle chamber and presented with stranger 1 in one compartment and an empty cage in another compartment for 10 min. The amount of time spent around each cage (stranger 1 or empty) was measured. At the end of the 10-min sociability trial, each subject mouse was then tested in a 10-min trial to quantitate social preference for a new stranger (social novelty preference trial). The wire cage enclosing the familiar C3H male mouse (stranger 1) was moved to the opposite side of the chamber that had been empty during the sociability trial. A second, unfamiliar C3H male mouse (stranger 2) was placed in the other side of the chamber in an identical small wire cage. The subject mouse was free to explore the mouse from the previous sociability test (stranger 1), and the novel mouse (stranger 2). The amount of time spent around each cage (stranger 1 or stranger 2) was measured. Data acquisition and analysis were performed automatically using Image CSI software (O'Hara & Co.; see 'Image analysis for behavioral tests').

## Contextual and cued fear conditioning test

The contextual and cued fear conditioning test was performed using a method similar to a previous report [72], with minor modifications. Fear conditioning was conducted in a transparent acrylic chamber (33 cm × 25 cm × 28 cm high) with a stainless-steel grid floor (0.2 cm-dimeter, spaced 0.5 cm apart; O'Hara & Co.). For the conditioning (conditioning test), each mouse was placed in the chamber and was allowed to explore freely for 3 min. Subsequently, white noise (55 dB) was played through a speaker set on top of the conditioning chamber wall, which served as the conditioning stimulus (CS), was presented for 20 s. During the last 2 s of CS presentation, mice received a footshock (0.7 mA, 2 s), which served as an unconditioned stimulus (US). Two more CS-US pairings were presented with a inter-stimulus interval of 40 s. Animals were returned to their home cages 40 s after the last CS-US paring. Twenty-four hours after conditioning (contextual test), contextual fear memory was tested for 3 min in the same chamber. Forty-eight hours after conditioning (cued test), cued fear memory was tested with an altered context. Each mouse was placed in a triangular chamber (33 cm × 33 cm × 32 cm high) made of opaque white plastic and allowed to explore freely for 1 min. Subsequently, each mouse was given CS presentation for 3 min. In each session, percentage of time spent freezing was calculated automatically using Image FZ software (O'Hara & Co.; see 'Image analysis for behavioral tests').

Pain sensitivity was measured as a control experiment using the fear conditioning chamber apparatus in a manner similar to a previous study [40]. Following 2 min of habituation, mice were given footshocks of increasing strength ranging from 0.05 to 0.7 mA in a stepwise manner by 0.05 mA, with an intertrial interval of 30 s. We measured current thresholds for three reactions of mice to nociceptive shock: flinch, vocalization, and jump (vertical and horizontal). Scoring indicated the first shock intensity at which each pain reaction was detected.

## Forced swim test with pharmacological manipulation

Forced swim tests were performed following Porsolt's method with minor modifications [39,74]. The apparatus consisted of a transparent plastic cylinder (22 cm height; 12 cm

diameter) placed in a box (41 cm × 31 cm × 42 cm high; O'Hara & Co). The cylinder was filled with water (22 ± 1°C) up to a height of 10 cm. Each mouse was placed into the cylinder and activity was monitored for 5 min via a CCD camera mounted on the top of the box. The cylinder was refilled with clean water after each test. Image data acquisition and analysis were performed automatically using Image PS software (see 'Image analysis for behavioral tests').

With respect to drugs, saline (0.9% NaCl in $H_2O$) was used as a vehicle and for control injections. Drug concentration for injections were: 15 mg/kg of imipramine (097–06491; FUJI-FILM Wako Pure Chemical Corporation, Osaka, Japan), 10 mg/kg of fluoxetine (F132; Sigma-Aldrich, MO, USA), 30 mg/kg of desipramine (042–33931; FUJIFILM Wako Pure Chemical Corporation). Drug concentrations were chosen on the basis of previous studies for imipramine [54][75][76][77][78], fluoxetine [54][75][79][78], and desipramine [54][75][76][78][80]. Both vehicle and drug solutions were intraperitoneally administered. Sixty min after injection, mice were tested in the open field for 10 min with 5 lux illumination and subsequently subjected to a forced swim test for 5 min.

## Image analysis for behavioral tests

The application software used for the behavioral studies (Image OFCR, LD, EP, CSI, PS, and FZ) were based on the public domain NIH Image program (developed at the U.S. National Institutes of Health and available at http://rsb.info.nih.gov/nih-image/) and ImageJ program (http://rsb.info.nih.gov/ij/), which were modified for each test (available through O'Hara & Co.).

## Subcellular fraction and western blot analysis

Subcellular fractions were prepared following Carlin's method [81] with minor modifications. All processes were carried out at 4°C. Briefly, WT and GluD1-KO mice with a C57BL/6N background (8 to 12 weeks old) were decapitated after cervical dislocation, and the frontal cortex (defined as one third anterior part of the cerebral cortex) and hippocampus were immediately dissected and removed. Brain tissues were homogenized in homogenization buffer [320 mM sucrose and 5 mM EDTA, containing complete protease inhibitor cocktail tablet (Complete Mini; Roche, Mannheim, Germany)] and centrifuged at 1,000 × g for 10 min. The supernatant was centrifuged at 12,000 × g for 10 min, and the resultant pellet was re-suspended in homogenization buffer as the P2 fraction. The P2 fraction was layered over a 1.2 M/0.8 M sucrose gradient and centrifuged at 90,000 × g for 2 h. The synaptosome fraction was collected from the interface, mixed with equal volume of Triton solution [1% Triton X-100, 0.32 M sucrose, 12 mM Tris-Cl (pH 8.0)] for 15 min, and centrifuged at 200,000 × g for 1 h. The resultant pellet was suspended in 40 mM Tris-Cl (pH 8.0), 1% SDS as the post synaptic density (PSD) fraction. The protein concentration was determined using BCA Protein Assay Reagent (Thermo Fisher Scientific, MA, USA). Equal volume of SDS sample buffer [125 mM Tris-Cl (pH 6.8), 4% SDS, 20% glycerol, 0.002% BPB, 2% 2-mercaptoethanol] was added to the sample fractions and boiled for 5 min at 100°C.

Protein samples were separated by 8% SDS-PAGE and electrophoretically transferred to nitrocellulose membranes (GE Healthcare, NJ, USA). Both WT and GluD1-KO mice samples were blotted on the same membrane for quantification. Membranes were blocked with 5% skimmed milk in TBS-T [20 mM Tris-Cl (pH 7.6), 137 mM NaCl, 0.1% Tween 20] for 1 h, and incubated with each primary antibody (1 μg/ml) (Table 1) for 3–4 h and horseradish peroxidase-conjugated secondary antibody for 1 h. Between these incubation steps, membranes were washed three times with TBS-T for 30 min. Protein bands were visualized with an enhanced chemiluminescence (ECL) kit (GE Healthcare) using a luminescence image analyzer with an

**Table 1. Primary antibodies used in the present study.**

|        | Sequence (NCBI #)           | RRID        | Host | Specificity | Reference/Source              |
|--------|-----------------------------|-------------|------|-------------|-------------------------------|
| **GluA1** | 841–907 aa (X57497)      | AB_2571752  | Rb   | KO          | FI (GluA1-Rb-Af690)           |
| **GluA1** | 880–907 aa               | n/a         | Rb   | IB          | [82]                          |
| **GluA2** | 175–430 aa (NM_013540)   | AB_2113875  | Ms   |             | Millipore (MAB397)            |
| **GluN2A** | 1126–1408 aa            | AB_2571605  | Rb   | KO          | [83] FI (GluRe1C-Rb-Af542)    |
| **GluN2B** | 1301–1456 aa (D10651)   | AB_2571762  | Rb   | KO          | FI(GluRe2C-Rb-Af300)          |
| **GluK2** | 844–908 aa (P42269)      | n/a         | Rb   | KO          | [84] Synaptic systems (180 003) |
| **GluD1** | 895–932 aa (NM_008166)   | AB_2571757  | Rb   | KO          | [32] FI (GluD1C-Rb-Af1390)    |
| **GluD2** | 897–934 aa (D13266)      | AB_2571601  | Rb   | KO          | [70]                          |
| **PSD-95** | 1–64 aa (D50621)        | AB_2571611  | Rb   | IB          | [85] FI (PSD-95-Rb-Af1720)    |

aa, amino acid residues; FI, Frontier Institute, Japan; GluA1, α-amino-3-hydroxy-5-methyl-4-isoxazole propionic acid (AMPA)-type glutamate receptor-1; GluA2, AMPA-type glutamate receptor-2; GluD1, δ-type glutamate receptor-1; GluD2, δ-type glutamate receptor-2; GluK2, kainate-type glutamate receptor-2; GluN2A, N-methyl-D-aspartate (NMDA)-type glutamate receptor 2A; GluN2B, NMDA-type glutamate receptor 2B; GP, guinea pig polyclonal antibody; KO, lack of immunohistochemical or immunoblot labeling in knockout mice; Ms, mouse monoclonal antibody; Rb, rabbit polyclonal antibody; RRID, Research Resource Identifier.

electronically cooled charge-coupled device camera (EZ capture MG; ATTO, Tokyo, Japan). Signal intensities of immunoreacted bands were determined using CS Analyzer ver.3.0 (ATTO).

## Statistical analysis

All data are expressed as mean ± SEM. Statistical analyses for behavioral studies were performed using EZR (Saitama Medical Center, Jichi Medical University, Saitama, Japan), which is a graphical user interface for R (The R Foundation for Statistical Computing, Vienna, Austria) [86]. Data were analyzed by one-way ANOVA, two-way ANOVA, two-way repeated-measures ANOVA followed by Dunnett's post hoc tests, or Student's t-test with Welch's correction as appropriate to correct for multiple comparisons. Attack latency in the resident-intruder test was analyzed using Kaplan-Meier survival curves followed by Mantel-Cox log-rank tests. All statistical tests were two-tailed. The level of significance set was $p < 0.05$.

## Results

### Locomotor activity and anxiety-related behavior in GluD1-KO mice

To determine whether GluD1 was involved in anxiety-related behavior, we performed tests in an open field (Fig 1A), an elevated plus maze (Fig 1F), and a light-dark transition test (Fig 1M). It is well established that performance in both the open field and elevated plus maze are influenced by the arena illumination levels [87,88]. We therefore used two different illumination conditions (5 and 100 lux) in these tests.

We performed the open field test for a total duration of 10 min (Fig 1A). The open field test presents a conflict between innate drives to explore a novel environment and personal safety [89]. GluD1-KO mice traveled longer distances compared to WT mice under both illumination conditions (Fig 1B and 1D). We also calculated the percentage of time spent in the central

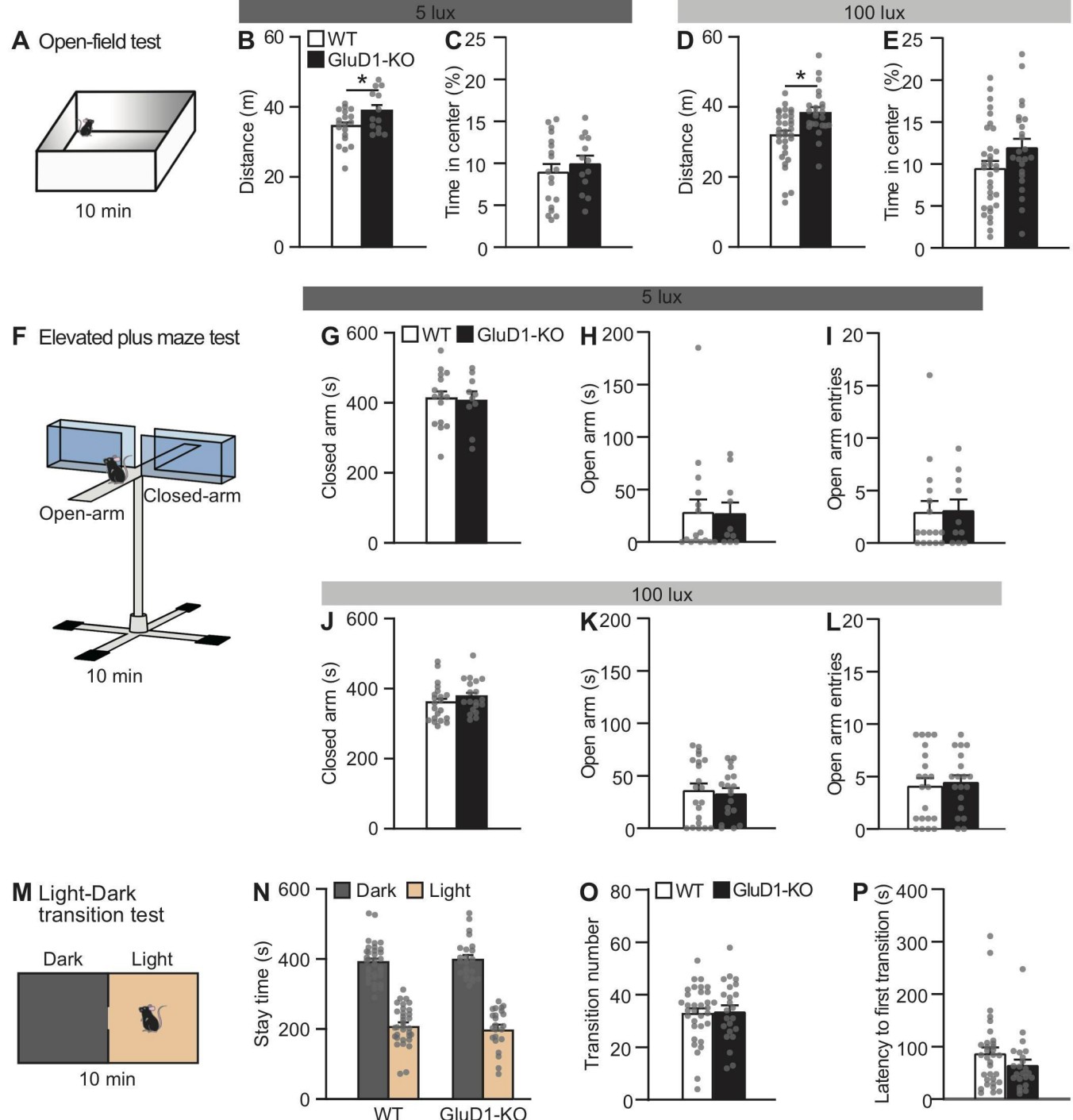

**Fig 1. Locomotor activity and anxiety-related behavior in GluD1-KO mice.** (A-E) The open field. Schematic representation of the open field test (A). GluD1-KO mice traveled significantly longer than WT in the open field test with 5 lux illumination (WT, n = 19; GluD1-KO, n = 13; p < 0.05, unpaired Student's t-test) (B), and 100 lux (WT, n = 33; GluD1-KO, n = 24; p < 0.05) (D). No significant difference was observed in the time spent in the central region with 5 lux (p = 0.46) (C) or 100 lux (p = 0.08) illumination (E). (F-L) The elevated plus maze. Schematic representation of the elevated plus maze (F). There were no significant differences between WT (5 lux, n = 16; 100 lux, n = 21) and GluD1-KO mice (5 lux, n = 10; 100 lux, n = 19) in the time spent in the closed arms [5 lux, p = 0.87 (G); 100 lux, p = 0.25 (J)] or in the open arms [5 lux, p = 0.95 (H); 100 lux, p = 0.72 (K)], or in the number of entries into the open arms [5 lux, p = 0.92 (I); 100 lux, p = 0.70 (L)]. (M-P) The light-dark transition test. Schematic representation of the light-dark transition test (M). There was no significant difference between WT (n = 33) and GluD1-KO mice (n = 23) in the time spent in the dark and light boxes [Dark box, p = 0.62; Light box, p = 0.62

(N)], in the number of entries into the light box (p = 0.85) (O), or in latency to first transition into the light box (p = 0.17) (P). *p < 0.05, unpaired Student's *t*-test with Welch'correction. All values presented are mean ± SEM.

area of the open field, which is commonly used as an index of anxiety [43]. There were no significant differences in the percentage of time spent in the central area between WT and GluD1-KO mice under either illumination conditions (Fig 1C and 1E).

In the elevated plus maze, we did not detect any significant differences in the time spent in the closed or open arms between genotypes, or in the number of entries into the open arm between genotypes under either of the illumination conditions (Fig 1G–1L). There was no significant differences between WT and GluD1-KO mice in total distance traveled using 5 lux (WT, 15.6 ± 1.0 meters; GluD1-KO, 18.3 ± 0.96 meters; p = 0.076) or 100 lux (WT, 19.5 ± 0.96 meters; GluD1-KO, 21.7 ± 1.1 meters; p = 0.142) and no differences in total entries using 5 lux (WT, 26 ± 2.4; GluD1-KO, 30 ± 2.8; p = 0.256) or 100 lux (WT, 34 ± 2.0; GluD1-KO, 39 ± 2.4; p = 0.162).

Besides, we performed another behavioral assay for studying anxiety in mice, the light-dark transition test (Fig 1M). There was no significant difference between WT and GluD1-KO mice in the time spent in the light and dark portions of the box (Fig 1N), in transition number between the illuminated and dark areas (Fig 1O), or in latency to enter the illuminated area of the box (Fig 1P).

Together, GluD1-KO mice showed higher locomotor activity in the open field test than the WT mice. However, GluD1-KO mice did not show any anxiety-related behaviors in the open field, elevated plus maze, or light-dark transition tests.

## Aggression-like behavior in GluD1-KO mice

GluD1-KO mice were rare to show aggressive-like behavior in their home cage. In accordance with these observations, there was no significant difference between groups, but a tendency for aggressive behavior in GluD1-KO mice in the resident-intruder test. (Fig 2).

## Sensorimotor gating in GluD1-KO mice

Because human mutations of the *Grid1* gene are associated with schizophrenia [23–25], we next performed PPI of the acoustic startle response, which is one of the most promising

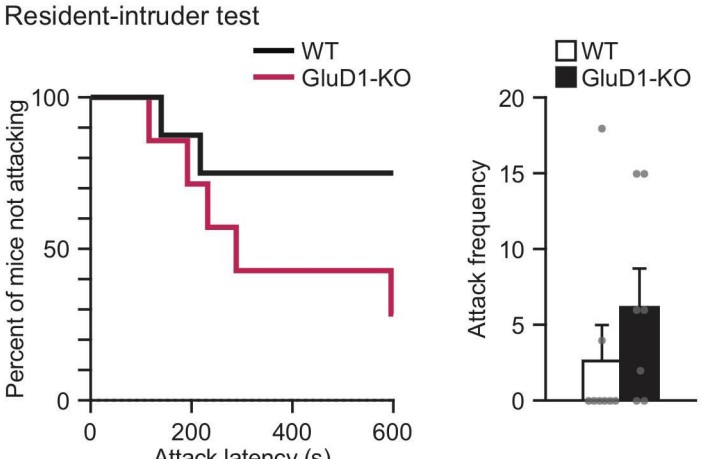

**Fig 2. Aggression-like behavior in GluD1-KO mice.** In the resident-intruder test, there was no significant difference between WT (n = 8) and GluD1-KO (n = 7) mice in attack latency (left; log-rank test, $\chi^2$ = 2.451, p = 0.117) or attack frequency (right, unpaired student's *t*-test, p = 0.304). All values presented are mean ± SEM.

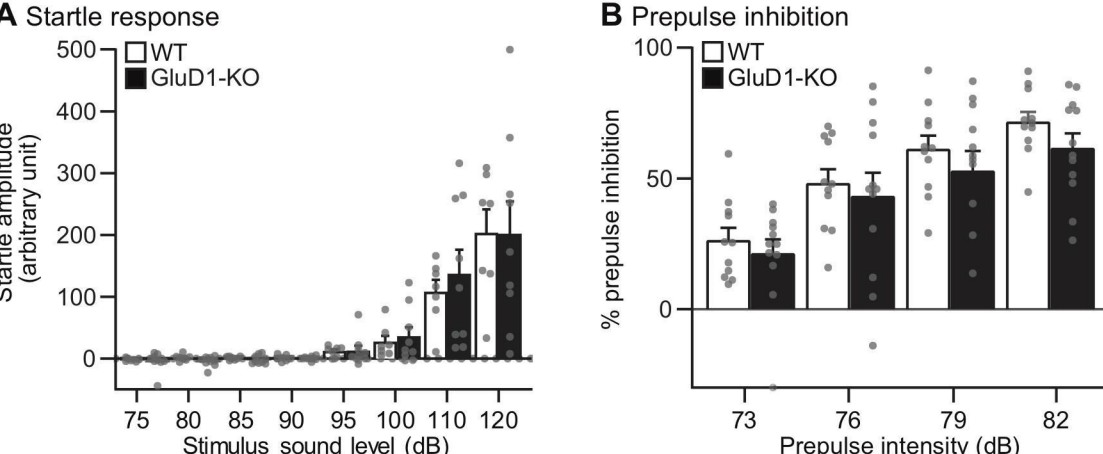

**Fig 3. Startle response and prepulse inhibition in GluD1-KO mice.** (A) Acoustic startle response test: Startle responses amplitudes were dependent on pulse intensity (WT, n = 7; GluD1-KO, n = 9) (two-way ANOVA: $F_{7,112}$ = 24.2, p < 0.001). There was no difference between genotype ($F_{1,112}$ = 0.08, p = 0.78), and no interaction between genotype and tone intensity ($F_{7,112}$ = 0.13, p = 1.00). (B) Prepulse inhibition test: The PPI levels of WT (n = 11) and GluD1-KO mice (n = 12) were not significantly different with prepulses of 73, 76, 79 and 82 dB (two-way ANOVA: Genotype; $F_{1,84}$ = 2.64, p = 0.11). PPI levels were dependent on prepulse intensity ($F_{3,84}$ = 18.09, p < 0.001). There was no significant interaction between genotype and prepulse intensity ($F_{3,84}$ = 0.08, p = 0.97). All values presented are mean ± SEM.

electrophysiological endophenotypes of both patients and animal models of schizophrenia [90–93]. In the acoustic startle responses, the amplitude of startle responses was dependent on pulse intensity. There was no difference between genotypes (Fig 3A).

We then examined PPI levels of WT and GluD1-KO mice using four different prepulse intensities. Induction of PPI using 73-, 76-, 79- and 82-dB prepulse in the 120-dB startle condition occurred in both WT and GluD1-KO mice (Fig 3B). There was no significant difference between WT and GluD1-KO mice in PPI levels, suggesting unchanged sensorimotor gating in GluD1-KO mice.

## Sociability and social novelty in GluD1-KO mice

We then examined the three-chamber social interaction test, which consists of a sociability test and a social novelty preference test [50] (Fig 4). In the sociability test, a wire cage with a stranger mouse (Stranger 1) was placed in one of the side chambers, and an empty cage was placed in another side chamber (Fig 4A). The preference of the mouse can be quantified based on the time spent around the wire cage with a stranger mouse versus the empty cage. GluD1-KO mice spent a significantly shorter time around the wire cage with the stranger mouse than that of WT mice (Fig 4B) indicating a lack of sociability.

In the social novelty preference test, a second stranger mouse (Stranger 2) was introduced into the empty cage. GluD1-KO mice spent a significantly shorter time around the wire cage with the novel stranger mouse (stranger 2) than that of WT mice (Fig 4C) indicating a lack of preference for social novelty.

## Contextual fear memories in GluD1-KO mice

To assess the involvement of GluD1 in fear memory, we performed contextual and cued fear conditioning tests (Fig 5). The freezing responses in the conditioning session did not differ significantly between genotypes (Fig 5A).

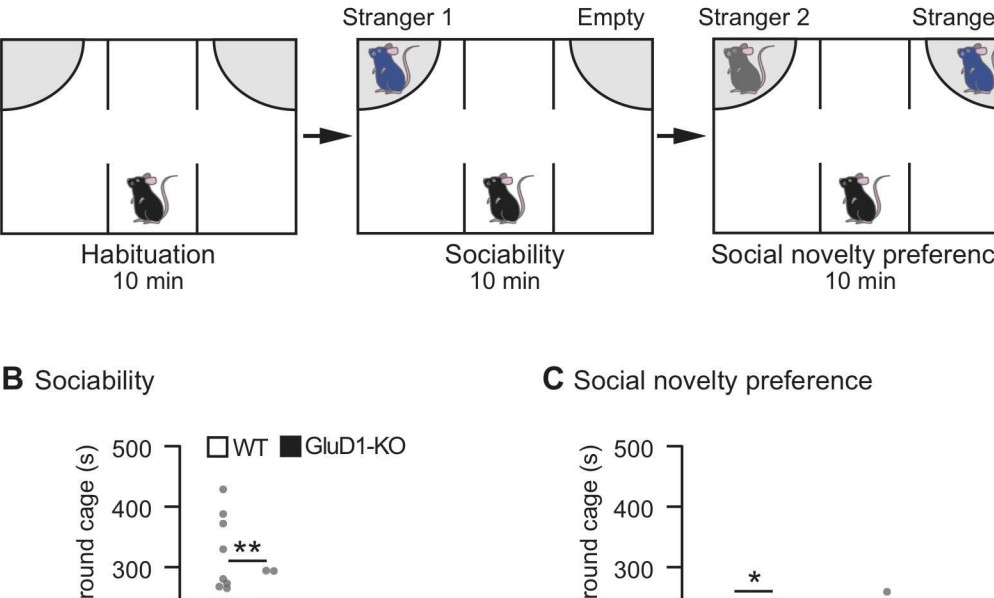

**Fig 4. Sociability and social novelty in GluD1-KO mice.** (A) Schematic representation of the three-chamber social interaction test. Sociability test (middle): a wire cage with a stranger mouse (Stranger 1) was placed in one side chamber and an empty wire cage was placed on the opposite side chamber. Social novelty preference test (right): a novel stranger mouse (Stranger 2) was placed in a wire cage in one side chamber and a familiar mouse (Stranger 1) was placed in a wire cage on the opposite site. (B) Sociability test: there was a significantly lower time spent near the wire cage with Stranger 1 in GluD1-KO (n = 23) than that of WT mice (n = 25) (p < 0.01). (C) Social novelty preference test: there was significantly lower time spent near the wire cage with stranger 2 in GluD1-KO mice than that of WT mice (p < 0.05). All values presented are mean ± SEM. *p < 0.05; **p < 0.01, unpaired Student's t-test with Welch's correction.

In the contextual test, GluD1-KO mice exhibited a modest but significant decrease in the freezing response relative to WT mice (Fig 5B). In contrast, GluD1-KO mice showed no significant difference, but a tendency towards less freezing response relative to WT mice in the cued test (Fig 5C). There were no significant differences in pain sensitivity (Fig 5D) or hearing ability (Fig 3A) between genotypes, suggesting that ablation of GluD1 caused deficits of contextual, but not cued memory in the fear conditioning tests.

## Depression-like behavior with pharmacological intervention in GluD1-KO mice

To analyze depression-like behavior, we used the Porsolt forced-swim test [74] (Fig 6). GluD1-KO mice showed significantly increased immobility, indicating enhanced depressive-like behavior (Fig 6A).

Next, we tested the impact of representative antidepressants on depression-like behavior in GluD1-KO mice. Imipramine and fluoxetine are inhibitors of the serotonin transporter, while desipramine is an inhibitor of the norepinephrine transporter. [94,95]. These drugs were injected intraperitoneally into WT or GluD1-KO mice 70 min before the forced-swim test (Fig

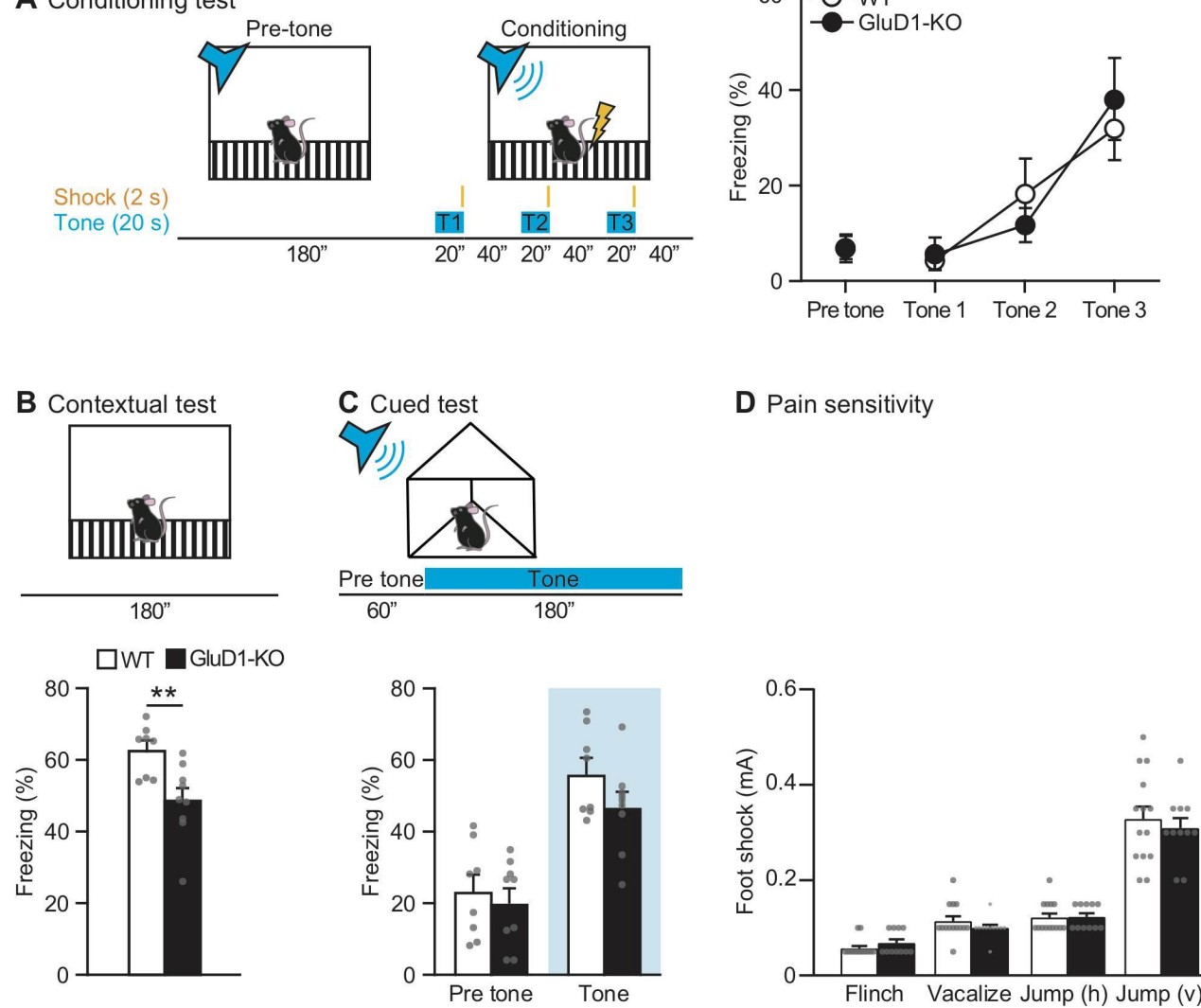

**Fig 5. Contextual and cued memory in GluD1-KO mice in the fear conditioning test.** (A) Schematic representation of the conditioning test (left). Freezing responses on the conditioning test; there was no significant difference between WT (n = 8) and GluD1-KO (n = 9) mice (two-way repeated-measures ANOVA: Genotype; $F_{1,15} = 0.0075$, p = 0.93, US presentation; $F_{3,45} = 14.5$, p < 0.0001, Genotype × US presentation; $F_{3,45} = 0.523$, p = 0.67). (B) Contextual test: freezing responses on the contextual testing 24 h after conditioning. There was significantly lower freezing in GluD1-KO mice during contextual conditioning (p < 0.01, unpaired Student's t-test). (C) Cued test: freezing responses on the cued testing 48 h after conditioning. There was no significant difference between WT and GluD1-KO mice in cued conditioning during pre-tone and tone (pre-tone, p = 0.62; tone, p = 0.14; unpaired Student's t-test). (D) Pain sensitivity test; there were no significant differences between WT (n = 14) and GluD1-KO (n = 11) mice in the footshock test evaluated by flinch (p = 0.24), vocalization (p = 0.24), vertical jump (p = 0.91) and horizontal jump (p = 0.56). All values presented are mean ± SEM. **p < 0.01.

6B). Before the forced-swim test, an open field test (10 min) was conducted to confirm that these drugs did not produce false-positive results on restoring depression-like behavior in the forced-swim test due to an increase of locomotor activity.

Injection of imipramine and desipramine, but not fluoxetine led to a reduction in the total distance in WT mice (One-way ANOVA: $F_{3,40} = 6.41$, p = 0.0012; Dunnett's *post hoc* test: Saline, 36.9 ± 2.3 meters; Imipramine, 25.5 ± 3.3 meters, p = 0.035; Fluoxetine, 37.0 ± 1.4 meters, p = 1.0; Desipramine, 27.5 ± 1.9 meters, p = 0.032) in the open field test. In contrast,

**A** Forced swim test

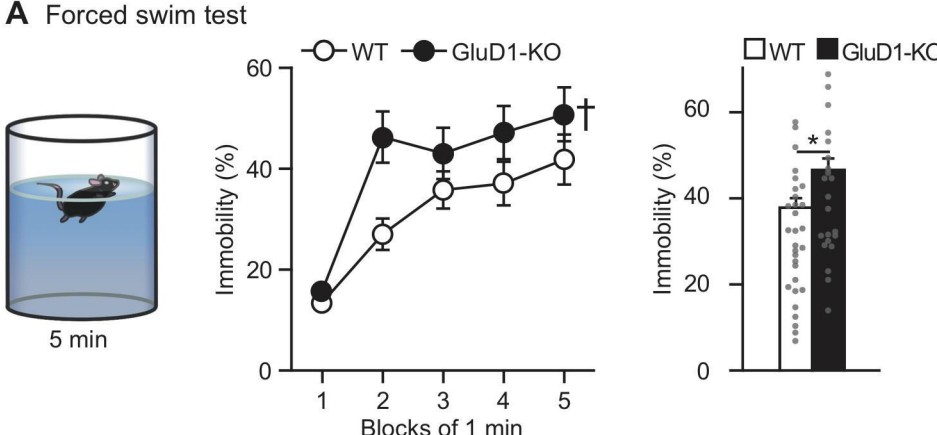

**B** Forced swim test with pharmacology

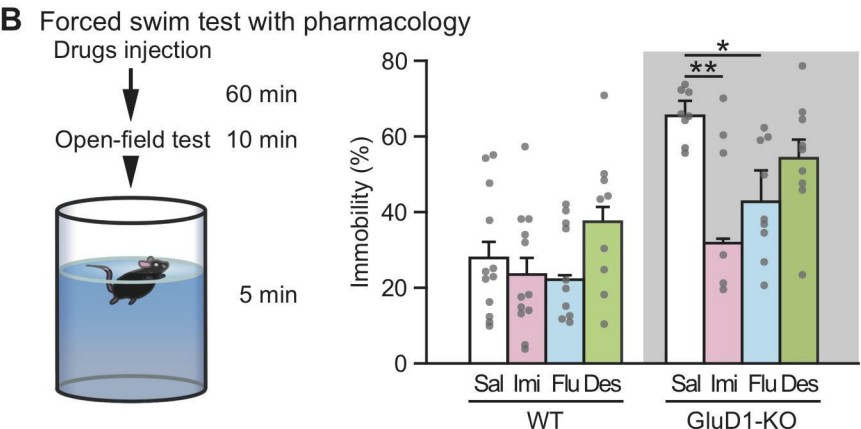

**Fig 6. Depressive-like behavior in GluD1-KO mice.** (A) Schematic representation of the forced-swim test (left). There was significantly higher percentage immobility in GluD1-KO mice (n = 22) than WT (n = 30) (middle) (two-way repeated measures ANOVA: Genotype, $F_{1,50}$ = 5.66, p < 0.05; Time, $F_{4,200}$ = 25.1, p < 0.001; Genotype × Time, $F_{4,200}$ = 1.56, p = 0.19). Average immobility times for minutes 1 to 5 in the forced-swim test (right). There was a significantly higher immobility in GluD1-KO than WT mice (p < 0.05, unpaired Student's t-test). (B) Impact of antidepressants on the forced-swim test. Animals were injected intraperitoneally with saline (WT, n = 12; GluD1-KO, n = 8), imipramine (15 mg/kg) (WT, n = 12; GluD1-KO, n = 8), fluoxetine (10 mg/kg) (WT, n = 11; GluD1-KO, n = 9), or desipramine (30 mg/kg) (WT, n = 9; GluD1-KO, n = 9). Mice were subjected to open field test 60 min after injection for 10 min, and subsequently subjected to a forced-swim test for 5 min. Average immobility times for 3 to 5 min: there was a significant genotype and treatment effect between WT and GluD1-KO mice (two-way ANOVA: Genotype, $F_{1,70}$ = 29.13, p < 0.001; Drug, $F_{3,70}$ = 6.16, p < 0.001), but no significant interaction (Genotype × Drug, $F_{3,70}$ = 2.56, p = 0.06). Within-genotype testing revealed that imipramine and fluoxetine led to a reduction in immobility in GluD1-KO mice (one-way ANOVA: $F_{3,30}$ = 5.58, p = 0.004; Dunnett's *post hoc* test (vs saline): Imipramine, p = 0.0017; Fluoxetine, p = 0.032; Desipramine, p = 0.42). In WT mice, no significant differences in immobility were observed with these antidepressants (one-way ANOVA: $F_{3,40}$ = 1.78, p = 0.17; Dunnett's *post hoc* test (vs saline): Imipramine, p = 0.85, Fluoxetine, p = 0.74; Desipramine, p = 0.41). All values presented are mean ± SEM. *p < 0.05; **p < 0.01, Dunnett's *post hoc* test. Desi, desipramine; Flu, fluoxetine; Imi, imipramine.

no significant differences were observed in these antidepressants in GluD1-KO mice ($F_{3,30}$ = 3.489, p = 0.028; Saline, 37.6 ± 2.0 meters; Imipramine, 31.7 ± 2.2 meters, p = 0.10; Fluoxetine, 38.1 ± 1.6 meters, p = 1.0; Desipramine, 31.8 ± 1.8 meters, p = 0.093).

In the forced-swim test, no significant differences were observed in these antidepressants in WT mice, possibly due to a ceiling effect. Of note, differences in percentage immobility for the saline-injected versus naïve WT mice in the forced-swim test may have arisen due to injection

stress. Injected of GluD1-KO mice with imipramine and fluoxetine, inhibitors of the serotonin transporter, led to a reduction in percentage immobility in the forced-swim test (Fig 6B). In contrast, a reduction in percentage immobility was not observed with GluD1-KO mice with desipramine, an inhibitor of the norepinephrine transporter (Fig 6B). These results suggest that inhibition of the serotonin transporter, but not norepinephrine transporter, restored depression-like behavior in GluD1-KO mice.

## Expression of glutamate receptors and PSD-95 in the frontal cortex and the hippocampus in GluD1-KO mice

Finally, we measured levels of synaptic protein expression in GluD1-KO mice. We analyzed the excitatory synaptic proteins, such as AMPA-type (GluA1 and GluA2), NMDA-type (GluN2A and GluN2B), Kainite-type (GluK2), and δ-type (GluD2) glutamate receptors and PSD-95 in the synaptosome and PSD fractions prepared from the frontal cortex and the hippocampus. Inconsistent with previous reports [40,41], we did not observe significant alterations of protein expression levels in any of these proteins in both synaptosome (Fig 7A) and PSD fractions (Fig 7B) in GluD1-KO mice. What did, however, find small but significant increases in expression of GluD2 in the PSD fractions of both the frontal cortex and the hippocampus of GluD1-KO mice (Fig 7B).

## Discussion

To avoid the variability inherent in mixed genetic backgrounds and effects of closely linked genes flanking the targeted locus, we generated GluD1-KO mice with a pure C57BL/6N background and performed behavior analysis to assess GluD1 functions *in vivo*. Our GluD1-KO mice showed a deficit in contextual, but not cued, fear memory, higher locomotor activity, abnormal social behavior, and enhancement of depressive-like behavior, that is partially consistent with the previous studies using *Grid1*[tm1Jnz] mice [39,40].

In the fear conditioning test, GluD1-KO mice showed significantly lower freezing times in 24-hr contextual tests. In contrast, GluD1-KO mice showed a trend toward an impairment of 48-hr memory in the cued fear conditioning; however, there was no significant difference between WT and GluD1-KO mice. It is well known that the hippocampus and amygdala are critical regions underlying contextual fear conditioning, whereas the amygdala underlies cued conditioning [8]. The deficit of GluD1-KO mice in the contextual test might suggest that GluD1 is more functionally important in the hippocampus. Interestingly, forebrain-specific knockout mice of Cbln1, a partner molecule of GluD1, showed a deficit in contextual and cued memory in the fear conditioning test [96]. The GluD1-CBLN2-NRXN transsynaptic adhesion system requires the formation and maintenance of synapses in the hippocampus *in vitro* and *ex vivo* [1,2]. Moreover, Cbln1 and Cbln2 double knockout mice have decreased synapse density in the hippocampus of 6-month-olds but not 1- or 2-month-olds [3]. Of note, we observed increased GluD2 expression in the PSD fraction of the hippocampus of GluD1-KO mice. In contrast, increased GluD1 expression has been reported in the cerebellum of GluD2-KO mice [9]. These results imply that compensatory regulation for GluD subunits expression exists in brain regions, including the hippocampus. Further analysis needs to clarify the specific role of the GluD-CBLN-NRXN transsynaptic adhesion system in the hippocampus. In addition, there are reports that GluD1 also has unique functions, such as (1) a regulator for group 5 mGluR-mediated AMPA-type glutamate receptor trafficking [4], and (2) the ion channel activity via group 1 mGluR signaling [5–7]. So, there are other possibilities that may explain how dysfunction of GluD1 affects hippocampal-dependent contextual fear memory. To determine details of the molecular function of GluD1, the hippocampus is a suitable brain

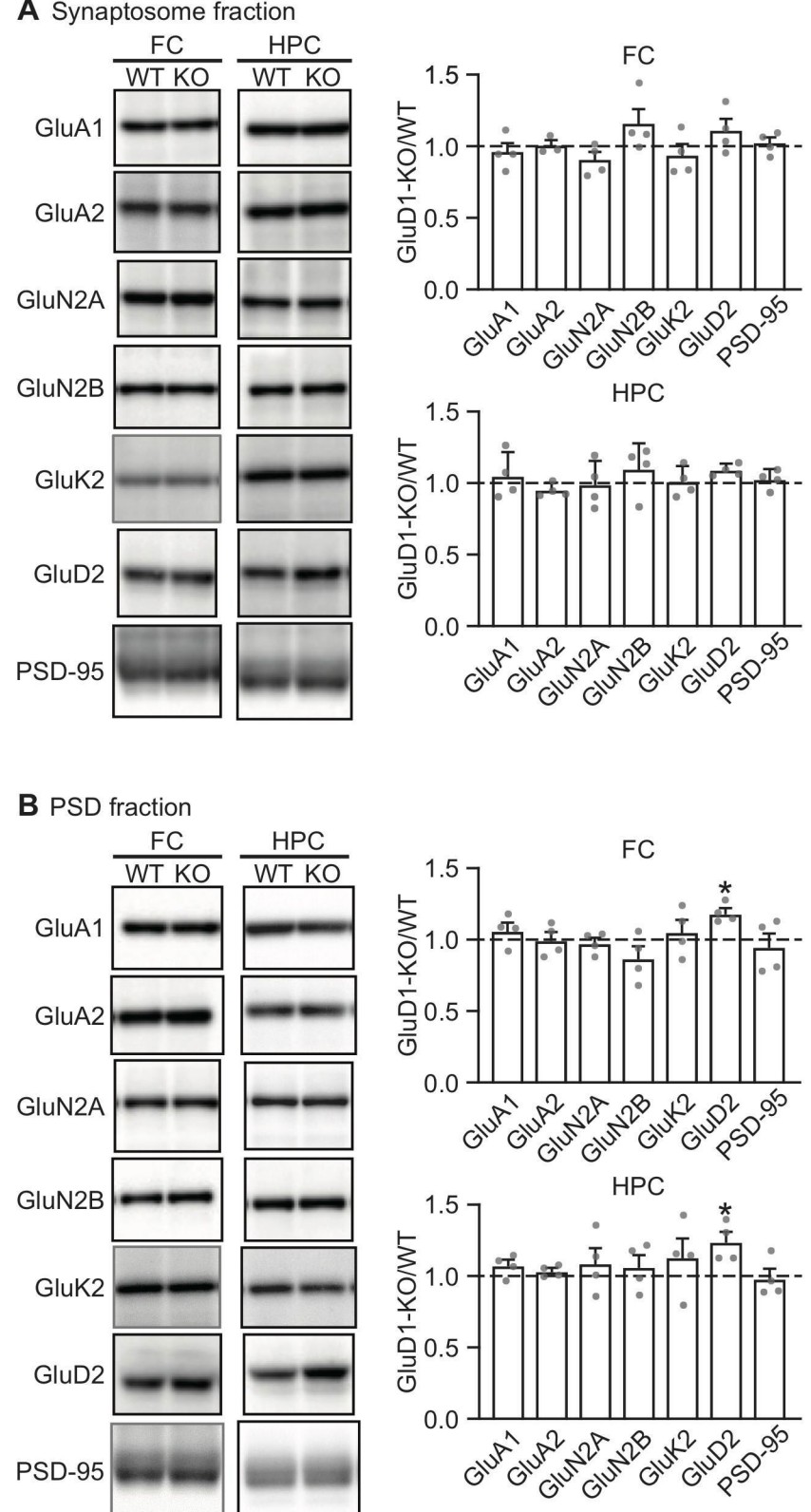

**Fig 7. Protein expression in synaptosome and PSD fractions prepared from the frontal cortex and the hippocampus of GluD1-KO mice.** (A) Protein expression in synaptosome fractions prepared from the frontal cortex

and the hippocampus. The protein loaded in lanes for GluA1, GluA2, GluN2A, GluN2B, GluK2, and PSD-95 were 20 µg, and 30 µg for GluD2. There was no significant difference between WT and GluD1-KO mice in the protein expression prepared from the synaptosome fractions of the frontal cortex (unpaired Student's t-test with Welch's correction: GluA1, p = 0.59; GluA2, p = 0.89; GluN2A, p = 0.34; GluN2B, p = 0.33; GluK2, p = 0.73; GluD2, p = 0.27; PSD-95, p = 0.71) and the hippocampus (GluA1, p = 0.59; GluA2, p = 0.60; GluN2A, p = 0.96; GluN2B, p = 0.46; GluK2, p = 0.86; GluD2, p = 0.22; PSD-95, p = 0.50). (B) Protein expression in PSD fractions prepared from the frontal cortex and the hippocampus. The protein loaded in lanes for GluA1, GluA2, GluN2A, GluN2B, GluK2, and PSD-95 were 10 µg, and 20 µg for GluD2. Except for GluD2, there was no significant difference between WT and GluD1-KO mice in protein expression prepared from the PSD fractions of the frontal cortex (GluA1, p = 0.50; GluA2, p = 0.99; GluN2A, p = 0.70; GluN2B, p = 0.43; GluK2, p = 0.71; GluD2, p = 0.01; PSD-95, p = 0.68) or the hippocampus (GluA1, p = 0.27; GluA2, p = 0.76; GluN2A, p = 0.51; GluN2B, p = 0.49; GluK2, p = 0.50; GluD2, p = 0.03; PSD-95, p = 0.89). All values presented are mean ± SEM from 3–4 experiments. *p < 0.05, unpaired Student's t-test with Welch's correction. FC, the frontal cortex; HPC, the hippocampus.

area for further analyses. Of note, the cued test was performed 24 h after conditioning in the *Grid1*[tm1Jnz] mice; in contrast, our GluD1-KO mice performed the context test 24 h subsequent to the cued test, that is, 48 h after conditioning. This somewhat different experimental procedure may have resulted in differences in the results. Further study is required to test the cued test 24 h after conditioning in our GluD1-KO mice.

Human GluD1 gene (*GRID1*) is a gene associated with susceptibility to schizophrenia, autism spectrum disorder, and depression [23–28,97,98]. However, how *GRID1* affects the pathophysiology of these conditions largely remains elusive. Our GluD1-KO mice showed significantly lower sociability in the three-chambered social interaction test, consistent with a previous report [39]. In contrast, our GluD1-KO mice showed lower social novelty preference that is inconsistent with a previous report [39]. Further analysis, such as novel object recognition, may clarify whether this phenotype is derived from either impairment of memory function, sociability, or both. In addition, further testing is required to check that the vision and olfaction in our GluD1-KO mice are intact, to confirm these abnormality did not affect the social deficiency.

Interestingly, lower *GRID1* mRNA expression is observed in the cerebral cortex of patients with schizophrenia [97] and autism spectrum disorder [99]. Downregulation of *GRID1* mRNA expression is also observed in iPS (induced pluripotent stem) cells derived from Rett syndrome patients, which is a condition associated with autism spectrum disorder [100]. In addition, downregulation of *Cbln1* mRNA was observed in mice carrying a triple dose of Ube3a, a model mouse for autism spectrum disorder [101]. Deletion of *Cbln1* in the glutamatergic neurons of the ventral tegmental area led to lower sociability by weakening excitatory synaptic transmission [101]. This behavioral abnormality might support the hypothesis that GluD1-CBLN-NRXN-dependent synapse formation and maintenance, in particular brain regions, is related to sociability.

Our GluD1-KO mice showed enhanced depressive-like behavior assessed by the forced-swim test. Pharmacological studies further revealed that imipramine and fluoxetine, but not desipramine, significantly restored the enhanced depressive-like behavior in GluD1-KO mice. Because imipramine and fluoxetine are more effective in inhibiting serotonin transporters than desipramine [94,95], increased serotonin concentration in the brain regions related to depressive-like behavior may account for the abnormal behavior of GluD1-KO mice; in other words, the serotonin signaling pathway might be altered in GluD1-KO mice. Candidate regions, in which GluD1 is expressed, related to depression are the lateral habenula and dorsal raphe nucleus [32,33,96]. Increased neuronal activity in the lateral habenula is observed in patients with depression and in animal models of depression [102–104]. In addition, lesions of the lateral habenula alters extracellular serotonin concentration in the dorsal raphe nucleus when receiving uncontrollable stress [105]. Furthermore, GluD1 mRNA is downregulated in

the frontal cortex of anhedonic rats as an animal model of depression [106], and this phenotype is completely reversed by intraperitoneal injection of the antipsychotic quetiapine [107]. Further analysis is required to identify regions involved in the enhanced depressive-like behavior mediated by the serotonergic system in GluD1-KO mice. As to the effect of antidepressants in the forced-swim test, selective serotonin reuptake inhibitors (SSRIs) tend to lead to more swimming behavior, whereas serotonin and norepinephrine reuptake inhibitors (SNRIs) tend to lead to more climbing behavior [108,109]. Thus, evaluating climbing and swimming behavior with more specific SSRIs (e.g. citalopram or escitalopram) [94,110] and SNRIs (e.g. reboxetine or atomoxetine) [111] will allow us to more precisely discriminate the pathway underlying enhanced depressive-like behavior in GluD1-KO mice. Lithium and D-cycloserine can rescue the depression-like behavior of *Grid1*[tm1Jnz] [39,41]. A future study examining whether the same effect is exhibited in our GluD1-KO mice is worthwhile.

We did not observe significant differences in (1) anxiety-related behavior evaluated by the open-field test, the light-dark transition test or the elevated plus maze, (2) aggressive behavior in the resident-intruder test, and (3) synaptic protein expression in our GluD1-KO mice compared to WT mice. In contrast, lower anxiety-related behavior, robust aggression, and alteration of synaptic proteins level were observed in *Grid1*[tm1Jnz] mice [39]. There are three conceivable possibilities that may explain the different behavioral phenotypes between our GluD1-KO and the *Grid1*[tm1Jnz] mice. The first possibility is the strain difference and flanking-genes effect. It is known that there are mouse-strain difference in basal levels of both anxiety [43–45,112] and aggression [46,47]. Moreover, it remains a concern that flanking alleles of the target locus generated during the backcrossing of ES cell-derived knockout mice to appropriate mice strains may influence phenotype [55,59,60]. The *Grid1*[tm1Jnz] mouse was generated using the 129S6/SvEvTac ES cell line [42] followed by backcrossing to C57BL/6 strain 2–6 times [39–41], so a phenotype difference may have arisen due to this. The second possibility is deleterious effects of selection maker gene in the target locus on the neighboring genes expression. The *Grid1*[tm1Jnz] mouse harbors the neomycin phosphotransferase cassette that allowed the selection of homologous recombinants in the targeted allele [42]. However, such marker genes can interfere with the transcription and splicing of the neighboring genes, thereby resulting in ambiguous genotype-phenotype relationships [66–69]. In addition, the gene encoding microRNA (miRNA) *miR-346* that regulates the translation of mRNAs via interaction with their 3' untranslated regions, is located in intron 2 of the GluD1 gene [97]. In contrast, the neomycin phosphotransferase cassette in our GluD1-KO mice deleted via the Cre/loxP system, so it is more appropriate for analyses of GluD1 function. The third possibility is a gene-environment interaction where differences in laboratory environments becomes an additional contributing factor that modulates the behavioral outcome of genetically modified animal models in psychiatry [113,114].

We also found that there was no statistically significant alteration of postsynaptic protein expression, including AMPA-, kainite-, and NMDA-type glutamate receptor subunits, and PSD-95 in the frontal cortex or the hippocampus of GluD1-KO mice, which is inconsistent with the previous studies [39–41]. In the prefrontal cortex, a significantly lower expression of GluA1 and GluA2 was observed in *Grid1*[tm1Jnz] mice [39]. In addition, there was a significantly lower expression level of GluA1, GluA2, and GluK2, and a significantly higher expression level of GluN2B and PSD-95 in the hippocampus of *Grid1*[tm1Jnz] mice [40]. Dramatic changes of synaptic protein expression levels in *Grid1*[tm1Jnz] may be due to their mixed genetic background. For example, it is well known that Disc1 is deficient in 129Sv/Ev strain [61,62] and it has been reported that increased GluN2A protein and NMDA current were observed following knockdown of Disc1 in primary cortical neurons [115]. Specific developmental disruption of Disc1 also reduced synaptic transmission in the hippocampus [116]. In addition, there is a copy

number variation between 129Sv and C57BL6: a large duplicated region in 129Sv strain on chromosome 14, which *Grid1* gene locates, and a region of copy-number gain in C57BL/6 strain on chromosome 7 [117]. Furthermore, gene expression profiling was different between 129SvEv and C57BL6 in the specific brain regions [118]. In the gene expression profiling between C57BL6J and 129S2 strain prepared from adult mouse brain, decreased NMDA-type (GluN2A and GluN2B) glutamate receptors and a higher amount of AMPA-type (GluA1 and GluA2) glutamate receptors, compared to C57BL/6J, were detected [119]. To eliminate such complexities, we would like to emphasize using genetically modified mice with a pure genetic background.

Behavioral analyses under a pure C57BL/6N genetic background suggest that GluD1 plays critical roles in contextual fear memory, sociability, and depressive-like behavior. We originally developed the *Grid1*$^{+/flox}$ mouse in which exon 4 of *Grid1* gene was flanked by loxP sequences [70]. *Grid1*$^{flox/flox}$ mice under a C57BL/6N genetic background allow us to delete the *Grid1* gene in a brain region-specific manner using region-specific Cre mice or Cre-expressing virus injections. This brain region-specific GluD1-KO mouse could be a useful tool to clarify the neuronal circuits and molecular mechanisms involved in contextual fear memory, sociability, and depressive-like behavior.

## Acknowledgments

We thank Akashi Kaori for technical assistance and advice; Moe Oono for drawing the graphical illustrations; Hisaaki Namba, and Hiroyuki Nawa for the provision and help with apparatus for the prepulse inhibition test; Hitoshi Uchida, Yu Ohmura, Shintaro Ohtsuka, Hidekazu Sotoyama and David Bett for insightful comments and suggestions.

## Author Contributions

**Conceptualization:** Chihiro Nakamoto, Manabu Abe, Kenji Sakimura.

**Formal analysis:** Chihiro Nakamoto, Tomonori Takeuchi.

**Funding acquisition:** Tomonori Takeuchi, Kenji Sakimura.

**Investigation:** Chihiro Nakamoto.

**Methodology:** Chihiro Nakamoto, Meiko Kawamura, Ena Nakatsukasa, Manabu Abe, Kenji Sakimura.

**Project administration:** Manabu Abe, Kenji Sakimura.

**Resources:** Rie Natsume, Masahiko Watanabe.

**Validation:** Chihiro Nakamoto, Keizo Takao, Tomonori Takeuchi.

**Visualization:** Chihiro Nakamoto, Tomonori Takeuchi.

**Writing – original draft:** Chihiro Nakamoto, Tomonori Takeuchi.

**Writing – review & editing:** Chihiro Nakamoto, Keizo Takao, Manabu Abe, Tomonori Takeuchi, Kenji Sakimura.

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
