## [Decision Letter · Decision Letter 0]

3 Dec 2019

PONE-D-19-30071

GluD1 knockout mice with a pure C57BL/6N background show impaired fear memory, social interaction, and enhanced depressive-like behavior

PLOS ONE

Dear Dr Takeuchi,

Thank you for submitting your manuscript to PLOS ONE. After careful consideration, we feel that it has merit but does not fully meet PLOS ONE’s publication criteria as it currently stands. Therefore, we invite you to submit a revised version of the manuscript that addresses the points raised during the review process.

Please respond to all of the reviewers' comments.

We would appreciate receiving your revised manuscript by Jan 17 2020 11:59PM. To enhance the reproducibility of your results, we recommend that if applicable you deposit your laboratory protocols in protocols.io, where a protocol can be assigned its own identifier (DOI) such that it can be cited independently in the future. For instructions see: http://journals.plos.org/plosone/s/submission-guidelines#loc-laboratory-protocols

We look forward to receiving your revised manuscript.

Kind regards,

Kazutaka Ikeda, Ph.D.

Academic Editor

PLOS ONE

Journal Requirements:

2. Please check the link for western blot images (10.6084/m9.figshare.10053092), as this does not appear functional.

3. Please see PLOS' recommendations for reporting research materials and useful Research Resource Identifiers (https://journals.plos.org/plosone/s/materials-and-software-sharing#loc-sharing-materials).

4. See PLOS' recommendations to ensure high standards of animal research reporting (https://journals.plos.org/plosone/s/animal-research#loc-reporting-guidelines) and ensure that you have reported your study in line with the ARRIVE guidelines.

Reviewers' comments:

Reviewer's Responses to Questions

**Comments to the Author**

1. Is the manuscript technically sound, and do the data support the conclusions?

Reviewer #1: Partly

Reviewer #2: Yes

2. Has the statistical analysis been performed appropriately and rigorously? 

Reviewer #1: Yes

Reviewer #2: No

3. Have the authors made all data underlying the findings in their manuscript fully available?

Reviewer #1: Yes

Reviewer #2: Yes

4. Is the manuscript presented in an intelligible fashion and written in standard English?

Reviewer #1: Yes

Reviewer #2: Yes

5. Review Comments to the Author

Reviewer #1: In this study, the authors characterized behavioral phenotypes of GluD1 KO mice that had a pure C57BL/6N genetic background. As reported previously for GluD1 KO mice (Grid-tm1Jnz) that had a mixed 129S6/SvEvTac; C57BL/6 genetic background, newly generated GluD1 KO mice showed hyper-locomotor activity, decreased sociability and social novelty preference, impaired contextual fear memory, and enhanced depressive-like behaviors. In contrast, a number of phenotypes were not observed in new GluD1 KO mouse lines. These include anxiety, aggression and sensory-motor gating, which were shown to be impaired in Grid-tm1Jnz GluD1 KO mice. While previous studies reported that lithium and D-cycloserine (DCS) rescued depressive-like behaviors of GluD1 KO mice, the authors now showed that the serotonin reuptake inhibitors were effective. Finally, although previous studies reported that synaptic proteins AMPAR (GluA1/2) and NMDAR (GluN2B) were decreased and increased, respectively, in the hippocampus and the cortex of GluD1 KO mice, the authors showed no such changes in new GluD1 KO mice.

It is a big concern in this research field that certain animal behaviors cannot be reproduced in different mouse lines in which the same genes were knocked out. Such inconsistent phenotypes could be caused by genetic backgrounds, remaining selection markers and laboratory environment. In this paper, the authors tried to clarify the phenotypes of GluD1 KO mice by generating them on a pure C57BL/6N genetic background. I think that the most important finding of this manuscript is that certain phenotypes are consistently observed in GluD1 KO mice on a pure C57BL/6N background despite that the previous Grid-tm1Jnz mouse line lacked the DISC1 gene and contained the Neo cassette. Thus, hyperactivity, reduced social interaction, impaired contextual fear memory and enhanced depression are likely the core phenotypes associated with the GluD1 gene. On the other hand, it remains incompletely clear whether and how inconsistent findings were observed between new GluD1 KO and Grid-tm1Jnz mouse lines as detailed below:

Major points:

1. Normal anxiety-like behavior (Fig. 1). In the open-field test, GluD1 KO mice showed a tendency to stay in the center (Fig. 1E). In addition, there is a large variation in the results of an elevated plus-maze test (Fig. 1H, I) and a light-dark transition test (Fig. 1P). The authors should increase N and show individual data.

2. Normal aggressive behavior (Fig. 2). There is a tendency that GluD1 KO mice showed more aggression. The authors should increase N and show individual data.

3. Normal cued fear conditioning (Fig. 5C). There is a tendency that GluD1 KO mice showed impaired cued fear conditioning. The authors should increase N and show individual data. To be consistent with the contextual fear memory test (Fig. 5B) and previous studies, the authors need to test cued fear memory 24 hours, not 48 hours, after conditioning.

4. The effect of the serotonin reuptake inhibitors on depression-like behavior (Fig. 6). This is an interesting new finding because lithium and DCS were previously shown to rescue the depression-like behavior. However, it remains unclear whether the effect of the serotonin reuptake inhibitor is specific to the new GluD1 KO mice on a pure C57BL/6 background. The authors should test the effect of lithium and DCS on their new GluD1 KO line.

5. Normal GluA and GluN2B levels (Fig. 7). It remains unclear whether this difference is caused by the genetic background or experimental conditions. The authors claim that previously reported changes may have been caused by normalization with actin levels. I wonder how such normalization procedures differentially caused changes in GluAs and GluN2B levels. At least the authors should directly test the effect of normalization on their samples.

6. To interpret the three-chambered social interaction test, the authors need to show that GluD1 KO mice had normal olfaction and vision.

Reviewer #2: Advanced Summary and Potential Significance to Field:

In their manuscript, Nakamoto et al. performed comprehensive behavioral phenotyping and biochemical analyses of the GluD1-knockout mice generated by the authors. The knockout (KO) mice were generated using the ES cell line RENKA, which is derived from pure C57BL/6N mouse strain and has a known genetic background. As the authors state, it is known that the genetic background of mutant mice alters their phenotypes. On the other hand, the importance of controlling the genetic background in mouse phenotyping analysis is not well known to researchers. Recently, ES cell lines derived from pure C57BL/6N strain have become available, but for the reasons described above, they are not commonly used in research. The Grid1 KO mice generated in the present study exhibited only some phenotypes in common with Grid1tm1Jnz mice used in some previous studies. However, the aggressive behavior observed in the resident-intruder paradigm and the enhanced anxiety in the open-field test and light/dark transition test exhibited by the Grid1tm1Jnz mice, were not observed in the KO mice in this present study. The Grid1tm1Jnz mice in the previous studies was generated using the 129S6/SvEvTac ES cells, which are known to carry mutations in several key genes involved in neurological disorders, such as Dsic1. Therefore, one of the reasons for the phenotypic differences maybe the genetic backgrounds of the ES cell lines used for the generation of the KO mouse. This study must be informative to the future generation of scientists and useful in the various disciplines of neuroscience. The authors have performed the experiments diligently and generated some very interesting results. However, I have several concerns:

Major points:

1. On page 5, line 110: The various sexual dimorphisms of the mouse phenotypes are known (1). Please clarify the reason why only the male mice were used.

(1) Karp et al. Prevalence of sexual dimorphism in mammalian phenotypic traits. Nat. commun. 8, 15475, 2017

2. On page 15, line 110: Please cite the corresponding literature for EZR.

3. In the manuscript, authors claimed that the KO mice showed “hyper locomotor activity.” However, the effect size of the gene mutation looks small. The "hyper" locomotor activity does not seem to be a suitable terminology for describing the phenotype. Again, have the authors examined locomotor activity in a familiar place for the animal, such as home cage? The anxiety induced by an unfamiliar place may have inhibited hyperlocomotion.

4. The number of subjects varies greatly among experiments (e.g., the number of wild-type animals exceeds 30 in the OF and is less than 10 in the RI). Please explain whether the different number of subjects affects the statistical power of the results.

5. On page 23, lines 500-507, the pharmacological tests using three drugs seem to be carried out independently. Therefore, the comparison by integrating all the subjects by using one-way ANOVA is not suitable in this case. Please consider comparing each control group with the corresponding drug-treated group by t-test or two-way ANOVA (effects of genotype and drug treatment). In addition, please consider adding the results visualized by graphs as supplemental data.

Minor points:

1. The authors should describe the official gene symbol of the mutated gene Grid1 in the abstract and the “key words” section.

2. On page 5, line 104, the name and supplier of the standard laboratory chow are lacking.

6. PLOS authors have the option to publish the peer review history of their article (what does this mean?). If published, this will include your full peer review and any attached files.

Reviewer #1: No

Reviewer #2: No

---

## [Author Response · Author response to Decision Letter 0]

16 Jan 2020

Review Comments to the Author

GluD1 knockout mice with a pure C57BL/6N background show impaired fear memory, social interaction, and enhanced depressive-like behavior

Chihiro Nakamoto, Meiko Kawamura, Ena Nakatsukasa, Rie Natsume, Keizo Takao, Masahiko Watanabe, Manabu Abe, Tomonori Takeuchi, Kenji Sakimura

Reviewer #1: In this study, the authors characterized behavioral phenotypes of GluD1 KO mice that had a pure C57BL/6N genetic background. As reported previously for GluD1 KO mice (Grid-tm1Jnz) that had a mixed 129S6/SvEvTac; C57BL/6 genetic background, newly generated GluD1 KO mice showed hyper-locomotor activity, decreased sociability and social novelty preference, impaired contextual fear memory, and enhanced depressive-like behaviors. In contrast, a number of phenotypes were not observed in new GluD1 KO mouse lines. These include anxiety, aggression and sensory-motor gating, which were shown to be impaired in Grid-tm1Jnz GluD1 KO mice. While previous studies reported that lithium and D-cycloserine (DCS) rescued depressive-like behaviors of GluD1 KO mice, the authors now showed that the serotonin reuptake inhibitors were effective. Finally, although previous studies reported that synaptic proteins AMPAR (GluA1/2) and NMDAR (GluN2B) were decreased and increased, respectively, in the hippocampus and the cortex of GluD1 KO mice, the authors showed no such changes in new GluD1 KO mice.

It is a big concern in this research field that certain animal behaviors cannot be reproduced in different mouse lines in which the same genes were knocked out. Such inconsistent phenotypes could be caused by genetic backgrounds, remaining selection markers and laboratory environment. In this paper, the authors tried to clarify the phenotypes of GluD1 KO mice by generating them on a pure C57BL/6N genetic background. I think that the most important finding of this manuscript is that certain phenotypes are consistently observed in GluD1 KO mice on a pure C57BL/6N background despite that the previous Grid-tm1Jnz mouse line lacked the DISC1 gene and contained the Neo cassette. Thus, hyperactivity, reduced social interaction, impaired contextual fear memory and enhanced depression are likely the core phenotypes associated with the GluD1 gene. On the other hand, it remains incompletely clear whether and how inconsistent findings were observed between new GluD1 KO and Grid-tm1Jnz mouse lines as detailed below:

Major points:

1. Normal anxiety-like behavior (Fig. 1). In the open-field test, GluD1 KO mice showed a tendency to stay in the center (Fig. 1E). In addition, there is a large variation in the results of an elevated plus-maze test (Fig. 1H, I) and a light-dark transition test (Fig. 1P). The authors should increase N and show individual data.

• First of all, we really appreciate that Referee 1 agrees with large concern about issues on replication of behavioral phenotype that are possible due to genetic background, remaining selection markers and the experimental condition in each laboratory.

• Secondly, we would like to emphasize that our main focus in this study was assessing the physiological function of GluD1 under pure C57BL/6N genetic background instead of comparing the phenotypes between our GluD1 and Gridtm1Jnz mice. To avoid making this kind of misunderstanding, we have re-structured our discussion. As such, our comparison with Gridtm1Jnz is moved to the end of the discussion (page 29, line 637 – page 31, line 683).

• Thirdly, according to Referee 1’s comment, we added individual data to all the figures. 

• Fourthly, Referee 1 pointed out that the number of animals were insufficient to resolve the concern on sample size. We would like to explain how to determine the number of animals in this paper. We estimated the sample size and effect size from the literature on Gridtm1Jnz mice [1–3] before commencing experiments. From these published results using Gridtm1Jnz mice, we expected the effect size to be high because there was a significant difference even when using less than 10 mice. The effect size was expected approximately d = 1-1.5. Therefore, we used 7-33 mice depending on the tests. We thought that the number of mice we used was in a range from previous reports in ‘Table A’ below. For these reasons, we think that we used a sufficient number of mice in this paper.

• We added in our methods how we decided the number of animals in this paper (page 5, line 114-115). 

• We cannot deny that our statistical power might not have been enough, so we replaced the pharase “normal anxiety-related behavior” to “Locomotor activity and anxiety-related behavior in GluD1-KO mice” in the results (page 15, line 327). We also replaced the phrase in the legend title of Figure 1 “Hyperlocomotor activity but normal anxiety-related behavior in GluD1-KO mice.” to “Locomotor activity and anxiety-related behavior in GluD1-KO mice.” (page 16, line 335).

Table A: Number of mice used in behavior analysis

Genotype (and/or strain, number of mice) [Reference]

• Open-field test

WT (N = 19), GluD1-KO (N = 13) [5lux; Present study]

WT (N = 33), GluD1-KO (N = 24) [100 lux; Present study]

WT (N = 8-22), GluD1-KO (N = 7-21) [2]

WT (129 strain, N = 11), WT (C57BL6 strain, N = 11) [4]

WT (C57BL6J, N = 20) [5]

WT (C57BL6J, N = 16), WT (129SvEvTac, N = 16) [6]

WT (N = 12), 5-hydroxytryptamine receptor 1B-KO (N = 10) [7]

WT (N = 32), Adenosine A2A receptor-KO (N = 28) [8]

WT (N = 19), MeCp2-mutant (N = 24) [9]

WT (N = 20), Neuroligin-4-KO (N = 18) [10]

• Light-dark transition test

WT (N = 33), GluD1-KO (N = 23) [Present study]

WT (C57BL6J, N = 10), Serotonin transporter-KO (C57BL6J, N = 10), WT (129SvEvTac, N = 12), Serotonin transporter-KO (129SvEvTac, N = 12) [11]

• Elevated plus maze test

WT (N = 16), GluD1-KO (N = 10) [5 lux; Present study]

WT (N = 21), GluD1-KO (N = 19) [100 lux; Present study]

WT (N = 7), GluD1-KO (N = 9) [2]

WT (C57BL6J, N = 10), Serotonin transporter-KO (C57BL6J, N = 10), WT (129SvEvTac, N = 12), Serotonin transporter-KO (129SvEvTac, N = 12) [11]

WT (C57BL6J, N = 16), WT (129SvEvTac, N = 16) [6]

WT (N = 20), Neuroligin-4-KO (N = 18) [10]

• Resident-intruder test

WT (N = 8), GluD1-KO (N = 7) [Present study]

WT (N = 9), GluD1-KO (N = 12) [2]

WT (N = 12), 5-hydroxytryptamine receptor 1B-KO (N = 14) [7]

WT (N = 17), TRP2-KO (N = 27) [12]

WT (N = 20), Adenosine A2A receptor-KO (N = 20) [8]

WT (N = 10), Neurokinin-1-KO (N = 10) [13]

WT (N = 9), MeCp2-mutant (N = 9) [9]

WT (N = 17-20), Serotonin transporter-KO (N = 17-20) [14]

WT (N = 19), Neuroligin-4-KO (N = 17) [10]

• Fear conditioning test

WT (N = 8), GluD1-KO (N = 9) [Present study]

WT (N = 6-8), GluD1-KO (N = 4-6) [1]

WT (C57BL6J, N = 9-11), WT (129SvEv/Tac, N = 9-11) [15]

WT (N = 20), Neuroligin-4-KO (N = 17) [10]

Control (flox/flox, N = 14), Forebrain specific-Cbln1-KO (N = 13)[16]

2. Normal aggressive behavior (Fig. 2). There is a tendency that GluD1 KO mice showed more aggression. The authors should increase N and show individual data.

• We have now added individual data to the figure. In the resident-intruder test, we expected a high effect size based on the Gridtm1Jnz study (WT = 9, Gridtm1Jnz = 12) [2], so we set the sample size (WT = 8, GluD1-KO = 7) to be almost the same as this previous paper. However, as pointed out by Referee 1, our GluD1-KO mice tended to be more aggressive than WT, but there was no statistical difference. Since we can not deny that our statistical power might not have been enough, we have rephrased the description of our results as follows: “In accordance with these observations, there was no significant difference between groups, but a tendency for aggressive behavior in GluD1 KO mice in the resident-intruder test.”(page 17, line 378-380). We also replaced the title of figure 2 “Normal aggression-like behavior in GluD1-KO mice.” to “Aggression-like behavior in GluD1-KO mice.” (page 18, line 382).

3. Normal cued fear conditioning (Fig. 5C). There is a tendency that GluD1 KO mice showed impaired cued fear conditioning. The authors should increase N and show individual data. To be consistent with the contextual fear memory test (Fig. 5B) and previous studies, the authors need to test cued fear memory 24 hours, not 48 hours, after conditioning.

• We added individual data to the graph in Fig 5C. In the fear conditioning test, we expected a high effect size based on the Gridtm1Jnz study (WT = 6-8, Gridtm1Jnz = 4-6) [1], so we set the sample size (WT = 8, GluD1-KO = 9) to be almost the same as in this previous paper. In this study, there was a tendency to decreased freezing of 48-hr cued fear conditioning test in GluD1-KO mice but was no significant difference compared to WT mice. As referee 1 mentioned, experimental condition was different between our study and the previous one: the previous study did a cued test 24 hr after conditioning and our study did 48 hr after conditioning test. In addition, our cued test was conducted 24 hr after the context test. So that it is hard to exclude the following possibilities: no significant difference of 48-hr cued test in GluD1-KO mice is due to either genetic background, low statistical power or experimental condition. Therefore, we have rephrased our descriptions as follows in our results: “In contrast, GluD1-KO mice showed no significant difference, but a tendency towards less freezing response relative to WT mice in the cued test” (page 20, line 442-443). And in our discussion: “In the fear conditioning test, GluD1-KO mice showed significantly lower freezing times in 24-hr contextual tests. In contrast, GluD1-KO mice showed a trend toward an impairment of 48-hr memory in the cued fear conditioning; however, there was no significant difference between WT and GluD1-KO mice.” (page 25, line 562 – page 26, line 565).

• However, the cued test at 24 hours is a very interesting experiment and we are planning to do the 24-hours cued test in the near future. We have added comments on this to our discussion (page 26, line 585-590).

4. The effect of the serotonin reuptake inhibitors on depression-like behavior (Fig. 6). This is an interesting new finding because lithium and DCS were previously shown to rescue the depression-like behavior. However, it remains unclear whether the effect of the serotonin reuptake inhibitor is specific to the new GluD1 KO mice on a pure C57BL/6 background. The authors should test the effect of lithium and DCS on their new GluD1 KO line.

• In this study, we focused on assessing the impact of representative antidepressants on depression-like behavior in GluD1-KO mice. However, we agree that it would be worthwhile to examine whether lithium and DCS could rescue enhanced depressive-like behavior in our GluD1-KO mice. We have added a sentence regarding lithium and DCS in our discussion (page 29, line 634-636).

5. Normal GluA and GluN2B levels (Fig. 7). It remains unclear whether this difference is caused by the genetic background or experimental conditions. The authors claim that previously reported changes may have been caused by normalization with actin levels. I wonder how such normalization procedures differentially caused changes in GluAs and GluN2B levels. At least the authors should directly test the effect of normalization on their samples.

• We appreciate a critical comment from Referee 1. We had realised that normalization by actin is less likely to explain the change in the expression level of GluA and GluN specifically in Gridtm1Jnz mice. We have re-considered the reason for the difference in the protein expression levels between Gridtm1Jnz and our GluD1-KO mice. One possibility is that dramatic changes of synaptic protein expression levels in Gridtm1Jnz may be due to mixed genetic background. On this point, we have rephrased the paragraph in our discussion (page 30, line 663-page 30, line 683).

6. To interpret the three-chambered social interaction test, the authors need to show that GluD1 KO mice had normal olfaction and vision.

• We appreciate essential suggestions from Referee 1. The previous study reported that Gridtm1Jnz mice had normal vision and olfaction, so we predicted our GluD1-KO has the same abilities. However, from our behavioral test, we cannot exclude the possibility that decreased social interaction in the GluD1-KO mice derived from the abnormality of vision and olfaction. So we add these possibilities in out discussion and plan to access olfaction and vision in GluD1-KO mice in future experiments (page 27, line 598-600). 

***

Reviewer #2: Advanced Summary and Potential Significance to Field:

In their manuscript, Nakamoto et al. performed comprehensive behavioral phenotyping and biochemical analyses of the GluD1-knockout mice generated by the authors. The knockout (KO) mice were generated using the ES cell line RENKA, which is derived from pure C57BL/6N mouse strain and has a known genetic background. As the authors state, it is known that the genetic background of mutant mice alters their phenotypes. On the other hand, the importance of controlling the genetic background in mouse phenotyping analysis is not well known to researchers. Recently, ES cell lines derived from pure C57BL/6N strain have become available, but for the reasons described above, they are not commonly used in research. The Grid1 KO mice generated in the present study exhibited only some phenotypes in common with Grid1tm1Jnz mice used in some previous studies. However, the aggressive behavior observed in the resident-intruder paradigm and the enhanced anxiety in the open-field test and light/dark transition test exhibited by the Grid1tm1Jnz mice were not observed in the KO mice in this present study. The Grid1tm1Jnz mice in the previous studies was generated using the 129S6/SvEvTac ES cells, which are known to carry mutations in several key genes involved in neurological disorders, such as Dsic1. Therefore, one of the reasons for the phenotypic differences maybe the genetic backgrounds of the ES cell lines used for the generation of the KO mouse. This study must be informative to the future generation of scientists and useful in the various disciplines of neuroscience. The authors have performed the experiments diligently and generated some very interesting results. However, I have several concerns:

Major points:

1. On page 5, line 110: The various sexual dimorphisms of the mouse phenotypes are known (1). Please clarify the reason why only the male mice were used.

(1) Karp et al. Prevalence of sexual dimorphism in mammalian phenotypic traits. Nat. commun. 8, 15475, 2017

• We used male mice because our aim is to evaluate the function of the GluD1 gene by using knockout mice. In this study, we prioritized minimizing the number of animals and thus, we only used male mice; female mice have an estrous cycle that influences behavior, such as anxiety-related behavior [19] and depressive-like behavior [20].

• However, as pointed out by Referee 2, there is a possibility that the phenotypes of GluD1-KO mice differs between males and females. Therefore, analysis of female GluD1-KO mice, and a comparison with males in warranted in the future.

2. On page 15, line 110: Please cite the corresponding literature for EZR.

• We have cited this corresponding literature for EZR in the method section (page 15, line 319).

Kanda Y. Investigation of the freely-available easy-to-use software “EZR” (Easy R) for medical statistics. Bone Marrow Transplant. 2013:48,452-458. advance online publication 3 December 2012; doi: 10.1038/bmt.2012.244

3. In the manuscript, authors claimed that the KO mice showed “hyper locomotor activity.” However, the effect size of the gene mutation looks small. The "hyper" locomotor activity does not seem to be a suitable terminology for describing the phenotype. Again, have the authors examined locomotor activity in a familiar place for the animal, such as home cage? The anxiety induced by an unfamiliar place may have inhibited hyperlocomotion.

• To address this comment, we have reconsidered and replaced the term “hyper-locomotor activity” with “increased locomotor activity” or “higher locomotor activity” in the open-field test (page 2, line 33; page 17, line 373; page 25, line 559). 

• Regretfully, we did not measure locomotor activity in the home cage which was another suitable parameter to see locomotor activity.

4. The number of subjects varies greatly among experiments (e.g., the number of wild-type animals exceeds 30 in the OF and is less than 10 in the RI). Please explain whether the different number of subjects affects the statistical power of the results.

• As we explained to Referee 1, we decided the sample size in this study based on previous studies of Gridtm1Jnz mice [1–3]. The effect size was expected to be high because there was a significant difference even when using less than 10 mice in these previous studies. In our behavior tests, such as anxiety, sociability, and depression-like behavior, we increased the cohort of mice group based on our estimated effect size in order to have sufficient statistical power. 

5. On page 23, lines 500-507, the pharmacological tests using three drugs seem to be carried out independently. Therefore, the comparison by integrating all the subjects by using one-way ANOVA is not suitable in this case. Please consider comparing each control group with the corresponding drug-treated group by t-test or two-way ANOVA (effects of genotype and drug treatment). In addition, please consider adding the results visualized by graphs as supplemental data.

• We completely agree with the comment on the statistical procedure; we actually had done this same analysis that Referee 2 suggested. We apologize for not having clearly presented this statistical analysis in the previous manuscript (Figure 6B, legend). So, we have rephrased this part to clarify this statistical analysis. We referred to the previous report [19]. We first confirmed there was a significant genotype effect, but no interaction between genotype and treatment in two-way ANOVA. We then performed one-way ANOVA with Dunnett's post hoc test for each genotype. The Dunnett's test is a method for testing the difference between the mean value of each of the control group and the treated group in the data of one control group (in this case, saline) and two or more treated groups. 

“Figure 6B (page 22, line 479): Mice were subjected to open field test 60 min after injection for 10 min, and subsequently subjected to a forced-swim test for 5 min. Average immobility times for 3 to 5 min: there was a significant genotype and treatment effect between and WT and GluD1-KO mice (two-way ANOVA: Genotype, F1,70 = 29.13, p < 0.001; Drug, F3,70 = 6.16, p < 0.001), but no significant interaction (F3,70 = 2.56, p = 0.06). Within-genotype testing revealed that imipramine and fluoxetine led to a reduction in immobility in GluD1-KO mice (one-way ANOVA with Dunnett’s post hoc test (vs saline): F3,30 = 5.58, p = 0.004; Imipramine, p = 0.0017; Fluoxetine, p = 0.032; Desipramine, p = 0.42). In WT mice, no significant differences in immobility were observed with these antidepressants (one-way ANOVA with Dunnett’s post hoc test (vs saline): F3,40 = 1.78, p = 0.17; Imipramine, p = 0.85, Fluoxetine, p = 0.74; Desipramine, p = 0.41). All values presented are mean ± SEM. *p < 0.05; **p < 0.01, Dunnett’s post hoc test. Desi, desipramine; Flu, fluoxetine; Imi, imipramine.”

• Due to this comment, we have shared raw data for all the behavioral analysis and western blot on figshare because we also believe that publishing raw data used in the graph is very useful, especially for future experimenters and meta-analysis.

• Raw data of behavior analysis:

https://doi.org/10.6084/m9.figshare.10052663.v1

• Raw data of western blot:

https://doi.org/10.6084/m9.figshare.10053092.v1

Minor points:

1. The authors should describe the official gene symbol of the mutated gene Grid1 in the abstract and the “key words” section.

• We added ’Grid1’ in the abstract and key words section (page 2, line 27).

2. On page 5, line 104, the name and supplier of the standard laboratory chow are lacking.

• We added the name and supplier of the laboratory chow (Oriental NMF, Oriental Yeast Co., Tokyo) (page 5, line 104-105).

***

References 

1. Yadav R, Hillman BG, Gupta SC, Suryavanshi P, Bhatt JM, Pavuluri R, et al. Deletion of Glutamate Delta-1 Receptor in Mouse Leads to Enhanced Working Memory and Deficit in Fear Conditioning. Christie B, editor. PLoS One. Public Library of Science; 2013;8: e60785. doi:10.1371/journal.pone.0060785

2. Yadav R, Gupta SC, Hillman BG, Bhatt JM, Stairs DJ, Dravid SM. Deletion of glutamate delta-1 receptor in mouse leads to aberrant emotional and social behaviors. Christie B, editor. PLoS One. Public Library of Science; 2012;7: e32969. doi:10.1371/journal.pone.0032969

3. Gupta SC, Yadav R, Pavuluri R, Morley BJ, Stairs DJ, Dravid SM. Essential role of GluD1 in dendritic spine development and GluN2B to GluN2A NMDAR subunit switch in the cortex and hippocampus reveals ability of GluN2B inhibition in correcting hyperconnectivity. Neuropharmacology. Elsevier Ltd; 2015;93: 274–284. doi:10.1016/j.neuropharm.2015.02.013

4. Gerlai R. Gene-targeting studies of mammalian behavior: is it the mutation or the background genotype? Trends Neurosci. Elsevier Current Trends; 1996;19: 177–181. doi:10.1016/S0166-2236(96)20020-7

5. Moy SS, Nadler JJ, Perez A, Barbaro RP, Johns JM, Magnuson TR, et al. Sociability and preference for social novelty in five inbred strains: an approach to assess autistic-like behavior in mice. Genes, Brain Behav. John Wiley & Sons, Ltd (10.1111); 2004;3: 287–302. doi:10.1111/j.1601-1848.2004.00076.x

6. Crabbe JC, Wahlsten D, Dudek BC. Genetics of Mouse Behavior: Interactions with Laboratory Environment. Science (80- ). 1999;284. Available: http://science.sciencemag.org/content/284/5420/1670.full

7. Saudou F, Amara D, Dierich A, LeMeur M, Ramboz S, Segu L, et al. Enhanced aggressive behavior in mice lacking 5-HT1B receptor. Science (80- ). 1994;265: 1875–1878. doi:10.1126/science.8091214

8. Ledent C, Vaugeois J-M, Schiffmann SN, Pedrazzini T, Yacoubi M El, Vanderhaeghen J-J, et al. Aggressiveness, hypoalgesia and high blood pressure in mice lacking the adenosine A2a receptor. Nature. Nature Publishing Group; 1997;388: 674–678. doi:10.1038/41771

9. Shahbazian M, Young J, Yuva-Paylor L, Spencer C, Antalffy B, Noebels J, et al. Mice with truncated MeCP2 recapitulate many Rett syndrome features and display hyperacetylation of histone H3. Neuron. Elsevier; 2002;35: 243–54. doi:10.1016/s0896-6273(02)00768-7

10. Jamain S, Radyushkin K, Hammerschmidt K, Granon S, Boretius S, Varoqueaux F, et al. Reduced social interaction and ultrasonic communication in a mouse model of monogenic heritable autism. Proc Natl Acad Sci U S A. National Academy of Sciences; 2008;105: 1710–5. doi:10.1073/pnas.0711555105

11. Holmes A, Li Q, Murphy DL, Gold E, Crawley JN. Abnormal anxiety-related behavior in serotonin transporter null mutant mice: the influence of genetic background. Genes, Brain Behav. John Wiley & Sons, Ltd (10.1111); 2003;2: 365–380. doi:10.1046/j.1601-1848.2003.00050.x

12. Stowers L, Holy TE, Meister M, Dulac C, Koentges G. Loss of Sex Discrimination and Male-Male Aggression in Mice Deficient for TRP2. Science (80- ). 2002;295: 1493–1500. doi:10.1126/science.1069259

13. Felipe C De, Herrero JF, O’Brien JA, Palmer JA, Doyle CA, Smith AJH, et al. Altered nociception, analgesia and aggression in mice lacking the receptor for substance P. Nature. Nature Publishing Group; 1998;392: 394–397. doi:10.1038/32904

14. Holmes A, Murphy D, Crawley J. Reduced aggression in mice lacking the serotonin transporter. Psychopharmacology (Berl). Springer; 2002;161: 160–167. doi:10.1007/s00213-002-1024-3

15. Balogh SA, Wehner JM. Inbred mouse strain differences in the establishment of long-term fear memory. Behav Brain Res. Elsevier; 2003;140: 97–106. doi:10.1016/S0166-4328(02)00279-6

16. Otsuka S, Konno K, Abe M, Motohashi J, Kohda K, Sakimura K, et al. Roles of Cbln1 in Non-Motor Functions of Mice. J Neurosci. Society for Neuroscience; 2016;36: 11801–11816. doi:10.1523/JNEUROSCI.0322-16.2016

17. Wei J, Graziane NM, Wang H, Zhong P, Wang Q, Liu W, et al. Regulation of N-methyl-D-aspartate receptors by disrupted-in-schizophrenia-1. Biol Psychiatry. NIH Public Access; 2014;75: 414–424. doi:10.1016/j.biopsych.2013.06.009

18. Li W, Zhou Y, Jentsch JD, Brown RAM, Tian X, Ehninger D, et al. Specific developmental disruption of disrupted-in-schizophrenia-1 function results in schizophrenia-related phenotypes in mice. Proc Natl Acad Sci U S A. National Academy of Sciences; 2007;104: 18280–5. doi:10.1073/pnas.0706900104

19. Marcondes FK, Miguel KJ, Melo LL, Spadari-Bratfisch RC. Estrous cycle influences the response of female rats in the elevated plus-maze test. Physiol Behav. Elsevier; 2001;74: 435–440. doi:10.1016/S0031-9384(01)00593-5

20. Meziane H, Ouagazzal A-M, Aubert L, Wietrzych M, Krezel W. Estrous cycle effects on behavior of C57BL/6J and BALB/cByJ female mice: implications for phenotyping strategies. Genes, Brain Behav. John Wiley & Sons, Ltd (10.1111); 2007;6: 192–200. doi:10.1111/j.1601-183X.2006.00249.x

---

## [Decision Letter · Decision Letter 1]

4 Feb 2020

GluD1 knockout mice with a pure C57BL/6N background show impaired fear memory, social interaction, and enhanced depressive-like behavior

PONE-D-19-30071R1

Dear Dr. Takeuchi,

We are pleased to inform you that your manuscript has been judged scientifically suitable for publication and will be formally accepted for publication once it complies with all outstanding technical requirements.

With kind regards,

Kazutaka Ikeda, Ph.D.

Academic Editor

PLOS ONE

Additional Editor Comments (optional):

Reviewers' comments:

Reviewer's Responses to Questions

**Comments to the Author**

1. If the authors have adequately addressed your comments raised in a previous round of review and you feel that this manuscript is now acceptable for publication, you may indicate that here to bypass the “Comments to the Author” section, enter your conflict of interest statement in the “Confidential to Editor” section, and submit your "Accept" recommendation.

Reviewer #1: All comments have been addressed

Reviewer #2: All comments have been addressed

2. Is the manuscript technically sound, and do the data support the conclusions?

Reviewer #1: Yes

Reviewer #2: Yes

3. Has the statistical analysis been performed appropriately and rigorously? 

Reviewer #1: Yes

Reviewer #2: Yes

4. Have the authors made all data underlying the findings in their manuscript fully available?

Reviewer #1: Yes

Reviewer #2: Yes

5. Is the manuscript presented in an intelligible fashion and written in standard English?

Reviewer #1: Yes

Reviewer #2: Yes

6. Review Comments to the Author

Reviewer #1: In response to my comments 1-3, the authors simply modified wording and admitted that they cannot rule out the possibility that their statistical power might not be enough to detect the phenotypes previously reported for Grid-tm1Jnz mouse line. I think that they could have made this paper much stronger and better if they repeated experiments to increase N. However, this paper has become solid and has its own value to show GluD1 KO phenotypes on a pure C57BL/6N background. So I would support its publication.

Reviewer #2: (No Response)

7. PLOS authors have the option to publish the peer review history of their article (what does this mean?). If published, this will include your full peer review and any attached files.

Reviewer #1: No

Reviewer #2: No

---

## [Editor Report · Acceptance letter]

12 Feb 2020

PONE-D-19-30071R1 

GluD1 knockout mice with a pure C57BL/6N background show impaired fear memory, social interaction, and enhanced depressive-like behavior 

Dear Dr. Takeuchi:

I am pleased to inform you that your manuscript has been deemed suitable for publication in PLOS ONE. Congratulations! Your manuscript is now with our production department. 

With kind regards,

on behalf of

Prof Kazutaka Ikeda 

Academic Editor

PLOS ONE